# A Model-Checking-Based Framework for Analyzing Ambient Assisted Living Solutions [note 1]

**DOI:** 10.3390/s19225057

**Published:** 2019-11-19

**Authors:** Ashalatha Kunnappilly, Raluca Marinescu, Cristina Seceleanu

**Affiliations:** 1School of Innovation, Design and Technology, Mälardalen University, 72220 Västerås, Sweden; cristina.seceleanu@mdh.se; 2Bombardier Transportation, 72223 Västerås, Sweden; raluca.marinescu@rail.bombardier.com

**Keywords:** ambient assisted living, architecture analysis and design language, statistical model checking, UPPAAL SMC

## Abstract

Since modern ambient assisted living solutions integrate a multitude of assisted-living functionalities, out of which some are safety critical, it is desirable that these systems are analyzed at their design stage to detect possible errors. To achieve this, one needs suitable architectures that support the seamless design of the integrated assisted-living functions, as well as capabilities for the formal modeling and analysis of the architecture. In this paper, we attempt to address this need, by proposing a generic integrated ambient assisted living system architecture, consisting of sensors, data collection, local and cloud processing schemes, and an intelligent decision support system, which can be easily extended to suit specific architecture categories. Our solution is customizable, therefore, we show three instantiations of the generic model, as simple, intermediate, and complex configurations, respectively, and show how to analyze the first and third categories by model checking. Our approach starts by specifying the architecture, using an architecture description language, in our case, the Architecture Analysis and Design Language, which can also account for the probabilistic behavior of such systems, and captures the possibility of component failure. To enable formal analysis, we describe the semantics of the simple and complex architectures within the framework of timed automata. We show that the simple architecture is amenable to exhaustive model checking by employing the UPPAAL tool, whereas for the complex architecture we resort to statistical model checking for scalability reasons. In this case, we apply the statistical extension of UPPAAL, namely UPPAAL SMC. Our work paves the way for the development of formally assured future ambient assisted living solutions.

## 1. Introduction

Elderly people across the world are offered enhanced care via the Ambient Assisted Living (AAL) solutions that support their independent and low-risk living. In order to facilitate the elderly efficiently and safely, it is often required that these solutions integrate various assisted-living functionalities like health monitoring, home monitoring, fall detection, robotic platform support, communication support, etc. Such integration is extremely beneficial in safety-critical situations, as in the following cases:A fall event occurring due to low pulse: In this case, if the fall sensor and the pulse monitoring sensor work independently of each other, no connection can be established between the two events, only an integrated solution would be able to indicate that the potential reason for the fall is in fact the person’s low pulse, which in turn may be critical for diagnosis (especially in case of patients having cardiac diseases).A high pulse detected during an exercise session: In case of such a scenario, the high pulse is absolutely normal, and hence no alarm should be raised. However, if the activity detection (in this case detecting an ’exercise session’) is not combined with pulse monitoring device, a false alarm will be triggered in the scenario.Simultaneous occurrence of fire and fall events: When both these events occur together, a safe mitigation of the scenario is achieved only when both these events are communicated to caregivers and firefighters, which is not guaranteed by independent systems working side by side. Assuming that the fire alarm communicated to the firefighters is verified for confirmation by a phone call to the user’s home, due to the inability of the elderly person to answer, the fire alarm may be deemed false and discarded, triggering a potential catastrophe [1,2].

Justified by the above, a timely integration of various assisted living functionalities is veridical. However, in literature, there are only few architectures, that address the concern of multiple-functionality integration in a timely and robust manner [1,2]. Due to their critical nature, it is beneficial that such behaviors (especially those emerging due to multiple functionality integration) are analyzed at early stages of development, for instance, at the design stage, using formal techniques, to provide some formal guarantees of meeting requirements. There has been some work in this direction, however, the existing frameworks [3,4] are still in infancy and cannot be used to specify the complete AAL system architecture including its artificial intelligent algorithms, timeliness, reliability, and fault-tolerance attributes.

In this paper, we address these shortcomings and propose an integrated architecture framework for describing AAL systems and a formal analysis framework that can be employed at the design stages of development. The integrated AAL architecture that we propose supports a range of assisted-living functionalities, like health monitoring, fall detection, reminder services, home monitoring, robotic platform support, etc., and follows the design of common AAL frameworks, with a variety of sensors, data collector unit, user interfaces, intelligent decision support system (DSS), local and cloud processing, etc. Our architecture gives due importance to intelligent decision making by proposing a DSS that employs a mix of artificial intelligent (AI) techniques, like fuzzy reasoning, rule-based reasoning (RBR) and case-based reasoning (CBR) for effectively modeling the context space and taking the respective actions based on the current context. The system architecture and its DSS are designed as a generic model that can be customized to fit various categories of architectures of different complexities. In this work, we show three such instantiations of our generic model, that is, (i) a minimal configuration that contains two sensors (pulse and fall), one user interface (a mobile phone), and a cloud controller with a simple DSS system to handle the events from both the sensors; (ii) an intermediate one with added sensors for blood pressure monitoring, motion detection and exercise monitoring and an enhanced cloud DSS; and (iii) a complex one comprising wider categories of health monitoring and home monitoring sensors, multiple user interfaces inclusive of robotic telepresence and vocal interactions, and a complex DSS system for handling multiple events simultaneously, and possessing both local and cloud copies for ensuring fault-tolerance via redundancy [5]. The system architecture, its DSS, and instance models are explained in detail in Section 4.

Our contributions also include a modeling and analysis framework proposed for the design-time analysis of complex AAL systems as described earlier. The architecture design relies on the Architecture Analysis and Design language (AADL) in which we show the structure and communication between the components of our proposed solution. In AADL, we are able to design the architecture together with the functional and error behavior of the constituting components (Section 2.1). Once described, the architecture needs to be analyzed formally to check if there are any functional errors and violations of quality-of-service attributes (end-to-end deadlines, fault tolerance, consistency, etc.). To enable this, we transform the architecture specifications into a formal model, in our case, the stochastic timed automata (STA) model, which can effectively capture the probabilistic behaviour of AAL components such as random component failures. We demonstrate our formal analysis via two techniques: (a) exhaustive model-checking using the state of the art model checker, UPPAAL [6], in the case of the minimal architecture configuration (for which exhaustive verification scales) and (b) statistical model-checking with UPPAAL SMC for analyzing the complex model instance [7]. The analysis results are described in Section 7 and also compared with the results obtained with another formal analysis tool, PRISM [8]. Our approach shows promising results of formally modeling and analyzing complex AAL system specifications, including fault-tolerant and AI-based decisions. A part of this work involving the complex instance of the architecture, its modeling and analysis are presented in the conference paper [5].)

The rest of the paper is organized as follows. In Section 2, we overview the basics of AADL, UPPAAL and UPPAAL SMC. Section 3 describes our proposed methodology. In Section 4, we describe our generic AAL system architecture and its instantiations. We present the AADL modeling constructs and the Agent Annex extension in Section 5. Section 6 describes the formal encoding of the AADL model, and in Section 7, we present the verification results applying the UPPAAL and UPPAAL SMC model checking on AAL system architectures; we also compare the results with those obtained with the PRISM model-checker. Related work is described in Section 9, and conclusions and future work are in Section 10.

## 2. Preliminaries

In this section, we briefly overview AADL, and the other formal notations and tools used for architecture analysis, that is, timed automata and stochastic timed automata, as well as UPPAAL and UPPAAL SMC.

### 2.1. The Architecture Analysis and Design Language

AADL [9] is a textual and graphical language in which one can model and analyze a real-time system’s hardware and software architecture as hierarchies of components at various levels of abstraction. AADL component categories like Application Software (Process, Data, Subprogram, Thread, Thread Group, etc.), Execution Platform (Device, Bus, Processor, Memory, etc.) and System are used to represent the run-time architecture of the system, however a more generalized representation is possible by specifying a component type as ***abstract*****.**

AADL allows possible component interactions via *ports/features*, *shared data*, *subprograms*, and *parameter connections*. In AADL, the input/output ports can be defined as: *event ports*, *data ports*, and *event-data ports*. Based on the component interactions, explicit *control flows*, and *data flows* can be defined across the interfaces of AADL components by specifying the components as *flow source*, *flow path*, or *flow sink*. The components can also be associated with various properties, like the *period* and *execution time* and the *dispatch protocol*. The *dispatch protocol* specifies if the component trigger is *periodic* or *aperiodic*.

A component in AADL can be defined by its *type* and *implementation*. The *component type* declaration defines the interface of the component (defining the component category and its interaction points with other components) and its externally observable attributes, whereas the *component implementation* defines its internal structure in terms of its subcomponents and connections between them. In this paper, we distinguish the subcomponents that are composed within a component in *port* interfaces in terms of their port interfaces. For instance, a *data component*, has no interfaces defined in terms of input-output ports, however it can be defined as a subcomponent of another component. We refer to such components as *Atomic Components*. However, if a component is composed of another component with port interfaces (like device, thread, abstract, etc.), then a well-defined component hierarchy is identified and we call such components as *Composite Components.*

The functional and error behavior of a component are described by the Behavior Annex (BA) [10] and the Error Annex (EA) [11] respectively, which model behaviors as transition systems. The BA state machine interacts with the component interface and represents the system behavior. Given finite sets of states and state variables, the behavior of a component is defined by a set of state transitions of the form $s\overset{guard,\;actions}{\to }s^{\prime }$, where *s*, $s^{\prime }$ are *states*, *guard* is a boolean condition on the values of state variables or presence of events/data in the component’s input ports, and *actions* are performed over the transition and may update state variables, or generate new outputs. Similarly, the EA models the error behavior of a component as transitions between states triggered by error events. It is also possible to represent the different types of errors, recovery paradigms, probability distribution associated with the error states and events, and also specify error flows and propagations within the component, and between various components.

In this paper, we focus on *abstract* components that allow us to defer from the run-time architecture of the system. The need for this generic model stems from the fact that in real-world applications like AAL, it is difficult to assign run-time semantics to components before the design matures. These generic component categories can be parametrized, and can be refined later in the design process through the “extends” capability of AADL. AADL allows us to archive these components and reuse them. For this, we partition them into two public packages in AADL, namely *component library* and *reference architecture* [12]. A *component library* creates a repository of component types and implementations with simple hierarchy. It can be established via two packages: (i) the *Interfaces Library* comprising generic components like sensors, actuators, and user-interfaces (UI); and (ii) the *Controller Library* that includes the control logic. The *reference architecture* creates a repository of components of complex hierarchy, e.g., the top-level system architecture.

### 2.2. Formal Notations and Tools

The formal analysis technique employed in this paper is *model checking*. We employ two different types of model checking in this paper: (1) *exhaustive model checking* using the state-of-the-art model checker UPPAAL [13]; and (2) *statistical model-checking*, using the statistical extension of UPPAAL model checker, UPPAAL SMC [7]. In the following, we overview the semantics of the input models and the mentioned tools.

#### 2.2.1. Timed Automata and Stochastic Timed Automata

A timed automaton (TA) as used in the model checker UPPAAL is a formal notation for describing real-time systems [14], and is defined by the following tuple:(1)$$TA=\langle L,l_{0},A,V,C,E,I\rangle$$
where *L* is a finite set of *locations*, $l_{0}\in L$ is the *initial location*, A=Σ∪τ is a set of *actions*, where Σ is a finite set of *synchronizing actions* (c! denotes the send action, and c? the receiving action) partitioned into inputs and outputs, $\Sigma =\Sigma_{i}\cup \Sigma_{o}$, and τ∉Σ denotes internal or empty actions without synchronization, *V* is a set of *data variables*, *C* is a set of *clocks*, $E\subseteq L\times B\left(C,V\right)\times A\times 2^{C}\times L$ is the set of *edges*, where B(C,V) is the set of *guards* over *C* and *V*, that is, conjunctive formulas of clock constraints (B(C)), of the form x⋈n or x−y⋈n, where x,y∈C, n∈N, ⋈∈{<,≤,=,≥,>}, and non-clock constraints over *V* (B(V)), and $I:L⟶B_{dc}\left(C\right)$ is a function that assigns *invariants* to locations, where $B_{dc}\left(C\right)\subseteq B\left(C\right)$ is the set of downward-closed clock constraints with ⋈∈{<,≤,=}. The invariants bound the time that can be spent in locations, hence ensuring progress of TA’s execution. An edge from location *l* to location $l^{\prime }$ is denoted by $l\overset{g,a,r}{\to }l$, where *g* is the guard of the edge, *a* is an update action, and *r* is the clock reset set, that is, the clocks that are set to 0 over the edge. A location can be marked as *urgent* (marked with an *U*) or *committed* (marked with a *C*) indicating that time cannot progress in such locations. The latter is more restrictive, indicating that the next edge to be transversed needs to start from a *committed* location.

The semantics of TA is a *labeled transition system*. The states of the labeled transition system are pairs (l,u), where l∈L is the current location, and $u\in R_{\geq 0}^{C}$ is the clock valuation in location *l*. The initial state is denoted by $\left(l_{0},u_{0}\right)$, where $\forall x\in C,\;u_{0}\left(x\right)=0$. Let u⊧g denote the clock value *u* that satisfies guard *g*. We use u+d to denote the time elapse where all the clock values have increased by *d*, for $d\in R_{\geq 0}$. There are two kinds of transitions:

(i) *Delay transitions*: $<l,u>\overset{d}{\to }<l,u+d>$ if u⊧I(l) and $\left(u+d^{\prime }\right)⊧I\left(l\right)$, for $0\leq d^{\prime }\leq d$, and

(ii) *Action transitions:*
$<l,u>\overset{a}{\to }<l^{\prime },u^{\prime }>$ if $l\overset{g,a,r}{\to }l^{\prime },a\in \Sigma ,u⊧g$, clock valuation $u^{\prime }$ in the target state $\left(l^{\prime },u^{\prime }\right)$ is derived from *u* by resetting all clocks in the reset set *r* of the edge, such that $u^{\prime }⊧I\left(l^{\prime }\right)$.

A stochastic timed automaton (STA) refines TA as follows: (i) probabilistic choices between multiple enabled transitions, where the output *probability* function γ may be defined by the user; and (ii) probability distributions for non-deterministic time delays, where the *delay density function*
μ is a uniform distribution for time-bounded delays or an exponential distribution with user-defined rates for cases of unbounded delays. Formally, an STA is defined by the tuple:(2)STA=〈TA,μ,γ〉.

The delay density function (μ) over delays in $R_{\geq 0}$ is either a uniform or an exponential distribution depending on whether the time in location *l* is bounded by an invariant, or is unbounded, respectively. With $E_{l}$ we denote the disjunction of guards *g* such that $l\overset{g,o,-}{\to }-\in E$ for some output *o*. Then d(l,v) denotes the infimum delay before the output is enabled, d(l,v)= inf $\left\{d\in R_{\geq 0}:v+d⊧E\left(l\right)\right\}$, whereas D(l,v)=sup $\left\{d\in R_{\geq 0}:v+d⊧I\left(l\right)\right\}$ is the supremum delay. If the supremum delay D(l,v)<∞, then the delay density function μ in a given state *s* is the same is a uniform distribution over the interval [d(l,v);D(l,v)]. Otherwise, when the upper bound on the delays out of *s* does not exist, $\mu_{s}$ is an exponential distribution with a rate P(l), where $P:L\to R_{\geq 0}$ is an additional distribution rate specified for the automaton. The output probability function $\gamma_{s}$ for every state s=(l,v)∈S is the uniform distribution over the set {o:(l,g,o,−,−)∈E∧v⊧g}.

In this paper, we use STA to model our AAL system architecture.

#### 2.2.2. UPPAAL and UPPAAL SMC

The UPPAAL model checker provides exhaustive model-checking of timed-automata models like the ones overviewed in Section 2.2. A real-time system can be modeled as a network of TA (NTA) composed via the parallel composition operator (“||”), which allows an individual automaton to carry out internal actions, while pairs of automata can perform handshake synchronization. The locations of all automata, together with the clock valuations, define the state of an NTA. The properties to be verified by model checking on the resulting NTA are specified in a decidable subset of (Timed) Computation Tree Logic ((T)CTL) [15], and checked by the UPPAAL model checker. UPPAAL supports verification of liveness and safety properties [13]. The queries that we verify in this paper are of the form: (i) **Reachability**: **E⋄p** means that there exists a path where *p* is satisfied by at least one state of the path; and (ii) **Time bounded leads to**: $p{⇝}_{\leq t}q$, which means that whenever *p* holds, *q* must hold within at most *t* time units thereafter.

UPPAAL SMC [7], the extension of UPPAAL for statistical model checking, provides the means to formally analyze stochastic models. A model in UPPAAL SMC consists of a network of interacting STA (NSTA) that communicate via broadcast channels and shared variables. In a broadcast synchronization one sender c! can synchronize with an arbitrary number of receivers c?. In the network, the automata repeatedly race against each other, that is, they independently and stochastically decide how much to delay before delivering the output, and what output to broadcast at that moment, with the “winner” being the component that chooses the minimum delay. In addition to the classical queries supported by UPPAAL, UPPAAL SMC also uses an extension of weighted metric temporal logic (WMTL) [16] to provide probability evaluation $Pr({*}_{x\leq C}\phi )$, where * stands for ⋄ (eventually) or □ (always), which calculates the probability that ϕ is satisfied within cost x≤C, but also hypothesis testing and probability comparison. In this paper, we will analyze only properties of the type “probability evaluation”.

## 3. A Framework for Formal Analysis of AAL Systems: Proposed Methodology

In this section, we present in detail the framework that we propose for modeling and verification of the AAL system architectures. We consider a generic architecture category for AAL systems that supports a variety of assisted living functionalities including health monitoring, home monitoring, fall detection, user interactions, and communication with family and caregivers.

Accordingly, the architecture supports a variety of components like sensors, a data collector unit to collect the sensor data, local and cloud processing, and intelligent decision support. The system architecture and its requirements are explained in detail in Section 4. This architecture design and the requirements in natural language form the input to our analysis framework. As depicted in Figure 1, the framework is composed of the following steps:
**Step** **1.**Create an abstract component-based model of the proposed architecture in AADL.

This step focuses on specifying the architecture using an architecture description language. In our case, we have chosen AADL [17] due to its rich semantics and suitability to model real-time embedded systems. In our approach, we demonstrate the modeling of AAL systems as abstract components and show how it can be extended to suit the specific instantiations (from simpler to more complex configurations, as shown in Section 4. The system modeling in AADL is presented in Section 5.
**Step** **2.**Define a semantic encoding of AADL model as an NSTA model.

Following the AADL modeling, in Step 2, we define the semantic anchoring of the AADL model as NSTA (Section 6). We present the semantic anchoring of the generic model and also show the above-mentioned instantiations of the latter to various configurations of increasing complexity. The NSTA model so formulated can be further analyzed via exhaustive model checking or statistical model-checking, depending upon the technique’s ability to cope with the model’s complexity. For the simple architecture configuration, we use exhaustive verification with UPPAAL and for the complex configuration, we use statistical model checking, using the tool UPPPAAL SMC. In the subsequent step, the functional and non- functional requirements of the architecture, which are initially specified in natural language are formalized as Timed Computation Tree Logic (TCTL) or Weighted Metric Temporal Logic (WMTL) queries to enable analysis in the NSTA model, using UPPAAL or UPPPAAL SMC. Consequently, Step 3 is formulated as follows:
**Step** **3.**Formalize the system requirements as queries expressed in the input language of the chosen model-checker.

As the final step, we verify the queries against the NSTA model of the architecture and gather the results (exact for UPPAAL and statistical for UPPAAL SMC) leading to Step 4 formulated as below:
**Step** **4.**Verify the queries in the model checker and gather verification results.

If the verification results show that requirements are not met, we feedback information from the verification (counter example or statistical information) to our design, which we modify and iterate steps 1, 2, 3, and 4.

## 4. A Generic AAL System Architecture

In this section, we detail the generic AAL system architecture that we propose. In addition, we also present the design of a novel decision support system for our system architecture that supports the integration of multiple functionalities and provides efficient decision making by combining multiple artificial-intelligent (AI) techniques as detailed later in this section. Finally, we present three specific instantiantions of the generic architecture model that follow the same modeling paradigms, yet which vary in their degree of complexity with respect to integrated functionalities.

The generic AAL system architecture is presented in Figure 2, and follows the architecture of many commercial AAL systems with various sensors, a data collector, DSS, security and privacy, database (DB) systems, user interfaces (UI), and cloud computing support. This architecture can act as a base for the development of many integrated AAL system architectures. We classify the sensors in the AAL environment as follows:Wearable sensors that send information as data (W_data), e.g., sensors measuring health parameters like pulse, ECG, etc. They are represented by the Sensor_A category in Figure 2;Non-wearable sensors measuring ambient parameters and health parameters (NW_data), e.g., camera sensors, motion sensors, etc., represented by the Sensor_B category;Wearable sensors that detect events (W_event), e.g., fall sensors, marked as the Sensor_C category;Non-wearable sensors detecting events (NW_event), e.g., fire sensors, denoted by the Sensor_D category.

A particular instantiation of the generic architecture can contain *n* sensors of each category, respectively, n∈N. As depicted in Figure 2, the data from the sensors are collected by the Data Collector unit, which processes the data by assigning labels and priorities. The Data Collector sends the data to the message queue in the Local Controller, where it gets sorted according to its priority such that when the DSS processes the first element in the queue, it processes the message with the highest priority. Our architecture has both local and cloud-based processing in order to ensure fault tolerance with respect to the DSS. The components of the architecture can interact via various communication protocols.

The crux of our AAL system is the **intelligent context-aware DSS**, shown in Figure 3. The novelty of our architecture stems from the combination of various AI algorithms, like *rule-based reasoning (RBR)*, *fuzzy logic*, and *case-based reasoning (CBR)* with context reasoning for efficient decision-making, as detailed below.

Our DSS architecture is inspired by the work of Zhou et al. [18], where the authors have proposed a context-aware, CBR-based ambient-intelligence system for AAL applications. CBR reasoning works very well in scenarios that are not specific and need to adapt accordingly to inputs. For instance, CBR reasoning is suited in a clinical decision support system that prescribes medicines/treatment, where the treatment, prescription, and medicine dosage vary for each patient, individually. CBR is an attractive choice due to its reasoning technique resembling more of human problem-solving competence, (i.e., trying to reason about a new scenario by looking at the similar solved cases in the past and adapting them according to the current needs) and less of knowledge engineering, however there are many scenarios that are specific and involve domain expertise, where RBR can be employed with more efficiency and ease.

For instance, if a fire occurs at home, the action to be taken by the system is to notify the firefighters, which can be easily implemented using *“if-then-else”* rules rather than via a CBR system that needs to compare across all cases using a case-matching algorithm to retrieve a matching case and act accordingly. Moreover, RBR systems using fuzzy logic are very efficient to determine sensor data deviations, if compared to crisp logic. For instance, the normal pulse range of a person is between 60–100 beats per minute, and a crisp rule-based-reasoning system (Boolean logic) classifies a pulse value of 59.5 or 100.5 beats per minute as an abnormal range (which in reality is not), consequently raising a pulse-deviation alarm to the caregiver. Using fuzzy logic, a degree of membership can be associated to each value, i.e., a pulse value of 59.5 or 100.5 is strictly not within abnormal or normal boundaries, rather it is considered 97% within normal range and 3% within abnormal range. Thus, by replacing the crisp boolean logic with fuzzy logic, a multitude of false pulse deviation alarms can be avoided. However, RBR (even fuzzy based) cannot work efficiently in many other ill-defined scenarios that require adaptability, like that of a clinical decision support system or a system that sends personalized recommendations to its users.

The DSS triggers the various AI algorithms based on a change in *context* [18]. The context-modeling (CM) and the usage of different AI algorithms are depicted in Figure 4.

As indicated, the CM module identifies the context space based on: (i) the personal profile of the user, e.g., gender, age, disease history, etc.; (ii) the activity of daily living (DA) performed by the user, e.g., exercising, sleeping, etc.; (iii) spatio-temporal properties, like time, location of the user, etc.; (iv) environmental, e.g., temperature, pressure, fire, etc.; and (v) health parameters, for instance, blood pressure (BP), pulse, blood glucose (BG), etc. Each of these context-space components can be associated with one of the three properties, *sensed, profiled*, or *predicted*. *Sensed* contexts are those directly derived from sensor values. *Predicted* contexts correspond to the output resulting from further analysis of sensed inputs, e.g., activity-recognition. *Profiled* values are usually descriptive and remain unchanged.

In our DSS, fuzzy reasoning is used for detecting DA [19], and also for determining sensor-data deviations (In order to reduce the complexity of our analysis, we have not explicitly modeled the DA detection using fuzzy logic and have often assumed that the user’s DA is known in various scenarios.). To take decisions in various situations, we employ RBR first, and CBR as second paradigm, i.e., upon a change in context, the RBR triggers first and checks if there exists a rule to handle that particular context, if not, it allows the CBR system to tackle the context based on its learning from previous scenarios. Developing an efficient case base, case matching and formulating the adaptation rules are the most complex aspects of a CBR system. In our system, each time an RBR outputs a rule, we save it as a *case* in the CBR system with the *case-id* represented by the DA of the user, the *context space* represented by the case features, and the triggered *rule* represented by the solution for a particular case. The Knowledge Base (KB) stores the context, rules, and cases. The internal structure of the DSS is represented in Figure 4. An example scenario of the DSS reasoning employing different AI techniques is presented in detail in Listing 1 of Section 6.2.

The generic architecture, and its DSS can be instantiated to create a family of AAL architectures that follows similar design principles. In this paper, we present three such architectures and their DSS instantiations.
**Category 1: A minimal configuration**—The minimum configuration architecture consists of the following modules: Two sensors (a fall sensor and a pulse monitoring sensor), a mobile phone UI, and cloud controller with a third-party UI and DSS system with a minimum context-space information including the health data (pulse and fall) and DA. The simplified DSS employs only RBR with fuzzy logic as AI techniques. The minimal configuration is shown in Figure 5.**Category 2: An intermediate configuration**—This instantiation (see Figure 6) is more complex than the previous one and it contains sensors belonging to all four types of the generic architecture (health monitoring sensors that detect pulse and blood pressure, smart home sensors that detect user movements, a wearable fall sensor, and a set of physical exercise monitoring sensors), as well as a local controller with inbuilt data collection functionality, which forwards the data to the cloud controller. The cloud controller has a DSS with context modeling, fuzzy logic, and RBR.**Category 3: A complex configuration**—In this category, we present the most complex version, the CAMI AAL architecture [2] derived from our generic model, and represented in Figure 7. The latter supports various *sensors* (e.g., a multitude of health and home monitoring sensors like the A&D UA-651 BLE blood pressure sensor [20], Fibaro temperature and motion sensor FGMS-001 [21], Fitbit bracelet [22], Vibby fall detection sensor [23], etc.), data collector, local controller (EXYS9200-SNG [24] referred as *CAMI gateway*), the *CAMI cloud*, and third party health platforms like *Open Tele* [25,26]. There is a set of user interfaces (UI) in CAMI, including robotic platforms (TIAGo [27] and Pepper [28]), mobile phone and vocal interface to facilitate the interaction with the elderly user. There is also a local backup of *DSS* in the CAMI gateway apart from the cloud. The communication between various modules can employ a variety of communication protocols, for instance, Bluetooth, Zigbee, Wifi, etc. The local processor is called the *CAMI gateway* and is responsible for all critical functionalities. The *Message Queue* is implemented by Rabbit MQ Message Broker [29]. The *DSS* is complex and employs context modeling, fuzzy logic, RBR and CBR. There are also redundant copies of DSS in the local controller and cloud controller.

In the following, we present the modeling and analysis of the simplest architecture (Category 1), by exhaustive model-checking as well as of the most complex one, the CAMI architecture (Category 3) by statistical model checking. We start by describing the use-case scenarios and system requirements of the two architecture instantiations, in the following section.

### 4.1. Use Case Scenarios and System Requirements

AAL systems should assist elderly users with a variety of health and home-related functions, as well as social inclusion ones. Let us assume the following critical scenarios where we can employ systems whose architectures conform to the ones of Categories 1 and 3 described above, respectively.


**Overall Scenario:**
*Jim is an elderly user living alone in his home. Jim suffers from chronic cardiac disease, slight memory loss, and falls frequently.*


If Jim uses the AAL system architecture of Category 1, the latter should assist in fulfilling the scenarios below:*Scenario 1—Assistance for detecting health parameter deviations:* Jim has sudden pulse variations detected by the pulse monitoring sensor, which is critical for cardiac patients. If the pulse is low, the DSS alerts the caregiver of a low pulse. If the pulse is high and the user is currently exercising, this is considered as normal, and if not, it sends an alert to the caregiver.*Scenario 2—Fall detection*: Jim falls heavily while exercising, the fall sensors detect the fall and the system immediately notifies the caregiver of the fall event.

However, if Jim needs additional functionality support, then he needs to acquire the CAMI AAL system (Category 3), which can handle additional scenarios to the already mentioned ones. The fall detection in CAMI is complex, as it employs a combination of wearable fall sensor (Vibby) and camera sensor for detecting the fall event.
*Scenario 3—Home-monitoring functionalities*: Jim forgets to switch off the cooker after cooking his dinner, which results in a fire in the house. The fire detection sensor of CAMI detects the fire and the system alerts the firefighters of the fire incident in Jim’s house.*Scenario 4—Combining various functionalities in case of multiple events occurring together*: Jim is cooking his breakfast. He suddenly feels dizzy and falls. The gas-based cooker is still on, and eventually starts a fire in Jim’s house. In this case, the CAMI system detects the simultaneously occurring events, and alerts the firefighter and caregiver of both the events. As a result, the firefighters and caregivers can immediately start the rescue without waiting for alarm confirmations, avoiding potentially dangerous consequences [1]. Further, if there are any health parameter variations detected for Jim along with the fall (for instance, a low pulse), the fall event can be associated with the low pulse, and the caregiver notified accordingly, which can help in further diagnosis.

All these scenarios are safety critical and have to be processed in real time. For architecture 1, we consider verifying the following requirements:

### 4.2. Requirements of the Minimal Architecture Model (Category 1):

**R1${}_{\mathrm{Arch}1}$**: If a high pulse is detected by the pulse sensor and the elderly user’s DA is not exercising, then the DSS sends a notification to caregiver within 20 s. This requirement relates to Scenario 1**R2${}_{\mathrm{Arch}1}$**: If a fall is detected by the fall sensor, then the DSS sends a notification to caregiver within 20 s. It is associated with Scenario 2.

### 4.3. Requirements of the CAMI Architecture (Category 3):

For the CAMI architecture, we consider verifying the following functional and quality-of-service (QoS) attributes, like fault tolerance and data consistency. Such verification is beneficial, as the system needs to be prototyped and the analysis offers some assessment of the system’s dependability.

**R1${}_{\mathrm{CAMI}}$**: If the fire sensor detects a fire, then the DSS sends a notification to the firefighters, within 20 s. This requirement corresponds to Scenario 3.**R2${}_{\mathrm{CAMI}}$**: If a fall is detected by the wearable or the camera sensor, then the DSS sends a notification to the caregiver, within 20 s. This requirement relates to Scenario 2.**R3${}_{\mathrm{CAMI}}$**: If fire and fall are detected simultaneously by the respective sensors, then the DSS should detect the presence of the simultaneous events and send notifications to both the firefighters and the caregiver indicating the presence of both events, within 20 s. This relates to Scenario 4.**R4${}_{\mathrm{CAMI}}$**: If there is a pulse data deviation indicating high pulse, the DA is “not exercising”, and the user has a disease history of a cardiac patient, then the DSS sends a notification to the caregiver, within 20 s. This relates to Scenario 1.**R5${}_{\mathrm{CAMI}}$**: The decisions taken by the local DSS are updated in the cloud DSS such that they are eventually synchronized. This requirement relates to the data-consistency requirement of CAMI.**R6${}_{\mathrm{CAMI}}$**: If the local DSS fails, then the cloud DSS eventually becomes active. It corresponds to the fault-tolerance aspect of the CAMI system.

The overall goal is to analyze the satisfaction of the above requirements by the respective architectures. We achieve this by first specifying the architectures in AADL, and then by semantically mapping the specification into a (network of) STA (N(STA)) that we model-check with UPPAAL (for architecture category 1) or statistically model-check with UPPAAL SMC (for CAMI).

## 5. System Modeling in AADL

The generic architecture, depicted in Figure 2 can be modeled in AADL as a set of interacting components. All the components are modeled as *abstract*, and can be easily extended to suit particular run-time representations appropriate for specific requirements.

In order to develop the AADL model, we classify the AADL components as:**Atomic Components (AC)**: Components that do not have hierarchy in terms of sub-components with port interfaces, but might contain sub-components without port interfaces.**Composite Components (CC)**: Hierarchical components that contain sub-components with and without interfaces. For example, data is a sub-component without interface and it can be part of an AC or CC hierarchy.

The system architecture itself can be considered a CC with other AC or CC as its sub-components. In order to encode the complex modeling aspects and facilitate the reasoning with functional behavior and errors, we propose a modeling format for both AC and CC as defined below.

### 5.1. AAL Atomic Components

An AC is defined by its component type, implementation, behaviour annex (BA), and error annex (EA). The component type definition specifies its name, category (i.e., “abstract”) and interfaces. We can also specify particular component properties and flows in the type definitions (While defining the component properties, we chose to include thread-related properties like the Dispatch Protocol, Component Execution Time etc., which later aid us in reasoning. All these thread-related properties need to be instantiated by a value and hence we chose it to be instantiated with some values specific to our architecture chosen. If the reader wishes to use the AADL model for a specific architecture of choice, we recommend to extend the abstract models and manually update the property values under consideration or add/delete properties.). The implementation of an AC defines the data sub-components. The AC’s BA has two states, *Waiting* and *Operational. Waiting*represents the initial state where the component waits for an input, and *Operational* represents the state to which a component switches upon receiving the input (if it has not failed). The AC’s EA uses four states to represent failure: *Failed Transient*, *LReset*, *Failed Permanent*, and *Failed ep*. The state *Failed Transient* models transient failures, from which a recovery is possible via a reset event. Since a reset is modeled as an internal event that occurs with respect to a probabilistic distribution, we model an additional location *LReset* to encode a component’s reset action upon the successful generation of the reset event. *Failed Permanent* models a permanent failure of the RBR, from which the component cannot recover. *Failed ep* models a failure due to error propagation from its predecessor components.

An example of an AC in the architecture is the RBR component of the CAMI DSS. In this paper, we illustrate the RBR for R3${}_{\mathrm{CAMI}}$ (Scenario 1), described in Section 2.1. The RBR component type, implementation, BA, and EA are shown in Listing 1. The component type definition specifies its name, category (i.e., “abstract”) and interfaces (Lines 2–15). The RBR component type describes that the component gets activated aperiodically, has an execution time of 1 s, and illustrates the data flows between the respective input and output ports. The implementation definition of RBR (Lines 17–20) defines the data sub-components like the fuzzy data output, personal information and daily activity of the user, which forms the context-space of Scenario 1.

In the BA (Lines 21–28), *Waiting* represents the initial state where the component waits for an input from the pulse sensor. In the *Operational* state, the system monitors the fuzzy logic output to identify any pulse variations. The fuzzy reasoning is not shown in Listing 1 as it is part of the context-reasoning module and not RBR, however we present the underlying reasoning in a nutshell. First of all, fuzzy data memberships are assigned to the range of pulse data values: Low [40–70], Normal [55–135], and High [110–300], where the numbers represent heart beats per minute. The pulse data inputs from the sensor are classified as Low, Normal, or High. If a high pulse is detected by the RBR, then the user context is tracked by checking the elderly’s activity of daily living and disease history. If the activity is “not exercising” and the user has a cardiac disease history, a notification alert is raised and sent to the caregiver. The information is encoded as a rule in the BA depicted in Listing 1. Upon triggering a particular rule, the RBR output is stored in the DB as a case input for CBR, where the case-id is represented by daily activity (DA), case features are the context space and the case solution is the RBR output (refer to Figure 4 to see the behavior of the various AI algorithms). The RBR output is also synchronized with Cloud DSS such that the data consistency is maintained. In the EA (Lines 30–49), we show the states - *Waiting* and *Failed Transient*, *Failed Permanent*, *LReset* and *Failed ep* plus their transitions based on a *TF* event (event that causes transient failures), *PF* (event that causes permanent failure) and *reset*event. If a *TF* or *PF* event occurs when the component starts, the latter moves to the *Failed Transient* state or *Failed Permanent* state respectively. From *Failed Transient*, the system can generate a reset event with occurrence probability of 0.9 and moves to *LReset*. If the recovery is successful with the reset event, the system moves to *Waiting* state with probability 0.8, else it moves to *Failed Permanent* with probability 0.2. In this work, we have considered the *Waiting* state in the EA and BA to be similar.

Listing 1: An excerpt from the RBR component in AADL for CAMI.

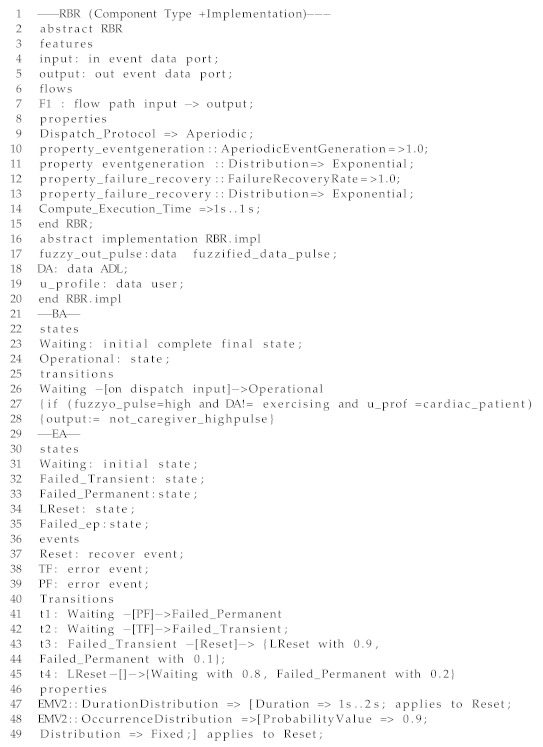



### 5.2. AAL Composite Components:

A CC is defined in a similar way as that of AC, except that its BA is not explicitly defined (we assume that the behaviour of the CC is already encoded by its sub-components). Also, the EA definition of CC shows the failure behaviour of its sub-components. In Listing 2, we present an excerpt of the DSS component, as an example of CC. The component type definition (Lines 2–12) is similar to that of an AC, except that we do not define explicitly properties like execution time of a CC (it is considered based on the execution time of each component, respectively). However, component implementation (Lines 13–26) shows the prototypes used to define sub-components and connections between them. The EA (Lines 28–39) shows the composite error behavior of DSS and shows that the DSS moves to *Failed Transient* or *Failed Permanent*, if each of its sub-components move to these states, respectively. No BA is created for the DSS since the behavior is defined by the BA of the sub-components.

Listing 2: An excerpt from the DSS component in AADL for CAMI.

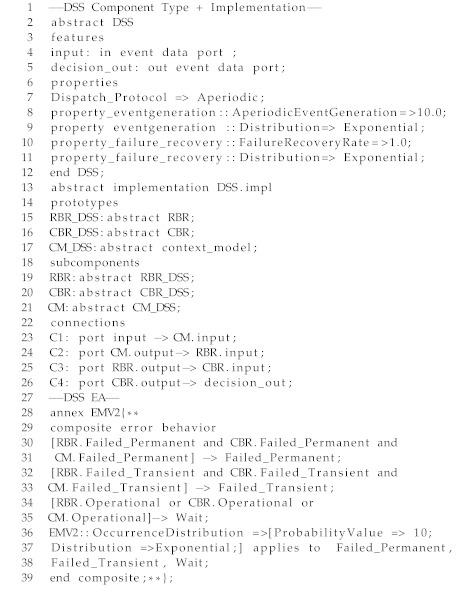



The assumptions made in the AADL model are: (i) all the system components have a reliability of 99.98%; (ii) the sensors have a periodic activation; (iii) all the system components interact via ports without any delay of communication; and (iv) the output is produced in the *Operational* state and there is no loss of information during transmission.

## 6. Semantics of AAL-Relevant AADL Components

AADL is a “semi-formal” language and in order to formally verify our AAL systems specified in AADL, we give formal semantics to AADL components (of the type used in this paper) in terms of *stochastic timed automata*, to be able to encode annex behaviors also. First, we provide the tuple definition of AADL components (Section 6.1), after which we perform a semantic anchoring of the AADL component tuple via a mapping between the elements of the AADL and the elements of the STA (Section 6.2).

### 6.1. Definition of AADL Components for AAL

An AADL component that we employ in this paper can be defined as a tuple:(3)$$AADL_{\mathrm{Comp}}=\langle Comp_{\mathrm{type}},Comp_{\mathrm{imp}},EA,BA\rangle ,$$
where $Comp_{\mathrm{type}}$ represents the component type, and $Comp_{\mathrm{imp}}$ represents the component implementation, BA the behavioral annex specification, and EA the error annex, as follows:$Comp_{\mathrm{type}}$ is defined as a tuple: $Comp_{\mathrm{type}}=\langle Features,Flow_{\mathrm{spec}},Prop\rangle$, where:
-Features = $IN_{p}$∪$OUT_{p}$, where $IN_{p}$, $OUT_{p}$ represent the sets of input
ports and output
ports respectively, and $IN_{p}$, $OUT_{p}$∈ {data-ports, event-ports, event-data-ports};-$Flow_{\mathrm{spec}}=\langle Flow_{\mathrm{so}},Flow_{p},Flow_{\mathrm{si}}\rangle$, where $Flow_{\mathrm{so}}$, $Flow_{p}$, $Flow_{\mathrm{si}}$ represent flow sources, flow paths and flow sinks respectively. Let $F_{s0}:Flow_{\mathrm{so}}\to OUT_{p}$ be a function that associates certain $OUT_{p}$ to $Flow_{\mathrm{so}}$ with $Flow_{\mathrm{so}}\subseteq OUT_{p}$, $F_{p}:Flow_{p}\to OUT_{p}\times IN_{p}$ be a function that associates and an input and an output to a flow, and $F_{\mathrm{si}}:Flow_{\mathrm{si}}\to IN_{p}$ be a function that associates certain $IN_{p}$ to $Flow_{\mathrm{si}}$, with $Flow_{\mathrm{si}}\subseteq IN_{p}$. For instance, in our AAL architecture, we can define $Flow_{\mathrm{spec}}$ for fall events by defining the output port of the fall sensor as $Flow_{\mathrm{so}}$, the input port of the cloud DSS as $Flow_{\mathrm{si}}$, and the input and output ports of all the intermediate components defining the Flow${}_{p}$;-Prop is the set of associated properties of the component, like Deployment, Communication, Timing, Thread-related
properties, etc. [12]. In this work, we only consider a subset of Timing, Thread-related
properties, and user-defined
properties, that are represented as follows: $Prop=\{T_{p},T_{e},Dispatch$
protocol,event_gen_dist,failure_recovery_dist} where $T_{p}$ and $T_{e}$ represent the period and execution-time of the component, respectively, $T_{p}$, $T_{e}$∈Timing
properties, Dispatch
protocol∈
{P,AP} (The dispatch protocol property of a thread determines when the thread is executed. A periodic thread is activated at time intervals of the specified period T; and an aperiodic thread is activated when an event arrives at a port of the thread.), where *P* represents a Periodic and AP represents an Aperiodic protocol, and P,AP∈
Thread-related
properties, and event_gen_dist,failure_recovery_dist∈
user- defined
properties represent the set of user-defined properties used for specifying the occurrence distribution of aperiodic events and failure recovery, respectively.
$Comp_{\mathrm{imp}}$ is defined as $Comp_{\mathrm{imp}}=\langle SC,P_{t},Con,MSM,Flow_{\mathrm{imp}},ETF\rangle$, where:
-SC represents the set of sub-components of the system with port interfaces ($SC_{i}$) and without port interfaces ($SC_{\mathrm{Data}}$), i.e., $SC=SC_{\mathrm{Data}}\cup SC_{i}$;-$P_{t}$ denotes the set of *Prototypes* used to define SC via $Fp:P_{t}\to SC_{i}\times SC_{\mathrm{Data}}$, a function that associates SC to a $P_{t}$, respectively;-Con represents the set of connections. $F_{\mathrm{con}}:Con\to Features$ is a function that assigns Features to Con;-MSM is the mode state machine that is modeled by a tuple, as follows: $MSM=\langle M_{s},\to \rangle$, where $M_{s}$ is the set of states, and $\to \subseteq M_{s}\times ev\times M_{s}$ is the transition relation (with ev being the set of events, such that Fe:
event-ports→ev, event-ports∈Features). We write $s\overset{e}{\to }s^{\prime }$ as short for $(s,e,s^{\prime })\in \to$, where $s,s^{\prime }\in Ms$, and e∈ev.The set of Con is defined with respect to MSM, if present;-$Flow_{\mathrm{imp}}$ are the flow implementations, represented as $Flow_{\mathrm{imp}}:SC\to Flow_{\mathrm{spec}}$;  ETF represents the set of end-to-end flows as complete flow paths from a starting $SC_{i}$ to the final $SC_{i}$, respectively.
The error annex EA is defined as the tuple: $EA=\langle E_{\mathrm{flows}},E_{\mathrm{beh}},E_{\mathrm{prop}}\rangle$, where:
-$E_{\mathrm{flows}}$ denotes the error flows, $E_{\mathrm{flows}}=\langle E_{\mathrm{pp}},Err_{\mathrm{so}},Err_{p},Err_{\mathrm{si}}\rangle$, where $E_{\mathrm{pp}}$ describes error propagations, and $Err_{\mathrm{so}}$, $Err_{p}$, $Err_{\mathrm{si}}$ represents error sources, error paths, and error sinks, respectively; $F_{e1}:Err_{\mathrm{so}}\to OUT_{p}$ is a function that associates certain output ports with error sources, $F_{e2}:Err_{p}\to \left(IN_{p},OUT_{p}\right)$ is a function that associates input and output ports via $Err_{p}$, $F_{e3}:Err_{\mathrm{si}}\to IN_{p}$ is a function that assigns certain input ports as error sinks;-$E_{\mathrm{beh}}$ represents error behavior, $E_{\mathrm{beh}}=\langle E_{s},\to_{e},E_{e},EM_{\mathrm{Comp}}\rangle$, where $E_{s}$ represents the set of error states, $\to_{e}$ denotes an error transition relation, $\to_{e}\subseteq$
$E_{s}\times$
$E_{e}\times$
$E_{s}$, with $E_{e}$, the set of error events. For a CC, the error behavior is represented as $EM_{\mathrm{Comp}}$ (error-model for a CC) with respect to the failure of its $SC_{i}$. Let $s_{e}$ and $s_{e}^{\prime }$ be two error states, $s_{e}$, $s_{e}^{\prime }\in E_{s}$, and $\to_{e}$ the transition between them due to an error event $e_{e}\in$
$E_{e}$, then $s_{e}{\overset{e_{e}}{\to }}_{e}\;s_{e}^{\prime }$.We represent the initial state as $s_{0e}\in E_{s}$. $F_{E_{\mathrm{pp}}}:Epp\to \left(IN_{p},OUT_{p}\right)$ is a function that associates input and output ports to error propagations;-$E_{\mathrm{prop}}$ denotes the error properties. In our work, we focus only on two error properties: *Duration distribution*($Dur_{\mathrm{dist}}$), and *Occurrence distribution* ($Occur_{\mathrm{dist}}$), which aid in our error analysis, thus $E_{\mathrm{prop}}=\left\{Dur_{\mathrm{dist}},Occur_{\mathrm{dist}}\right\}$.
The Behaviour Annex, BA is defined as: $BA=\langle B_{v},B_{s},\to_{b}\rangle$, where $B_{v}$, $B_{s}$, represent the set of variables, and the states of BA, respectively and $\to_{b}$ is a BA transition relation. Let $s_{b}$ and $s_{b}^{\prime }$ be two states of BA, $s_{b},s_{b}^{\prime }\in B_{s}$, and $\to_{b}$ the transition between them, $\to_{b}\subseteq B_{s}\times B_{v}\times SC_{\mathrm{Data}}\times B_{s}$, with $SC_{\mathrm{Data}}$ being the set of data subcomponents. We denote by $s_{0b}\in B_{s}$ the initial state of a BA path.

Formally, we distinguish the Atomic Component from the Composite Component as follows:AC∈$AADL_{\mathrm{Comp}}$, where $Comp_{\mathrm{ImplAC}}$= $\{SC_{\mathrm{Data}}$}, $EA_{\mathrm{AC}}\neq \emptyset$, where $E_{\mathrm{beh}}\in EA_{\mathrm{AC}}=\left\{E_{s},\to_{e},E_{e}\right\}$, $BA_{\mathrm{AC}}\neq \emptyset$,CC∈$AADL_{\mathrm{Comp}}$, where $Comp_{\mathrm{ImplCC}}$= $\left\{P_{t},SC_{i},SC_{\mathrm{Data}},Con,MSM,Flow_{\mathrm{imp}},ETF\right\}$, $EA_{\mathrm{CC}}\neq \emptyset$, where $E_{\mathrm{beh}}\in EA_{\mathrm{CC}}=\left\{EM_{\mathrm{Comp}}\right\}$, $BA_{\mathrm{CC}}=\emptyset$. A CC represents the system-level view of the architecture.

Next, we present an instantiated example of an AC and a CC from the CAMI architecture. The RBR component of DSS is an AC and it is defined by its type, implementation, BA, and EA (Listing 1). In formal semantics, we define it as follows:(4)$$RBR_{\mathrm{AADL}}=\langle Comp_{\mathrm{type}}\;\mathrm{RBR},Comp_{\mathrm{imp}}\;\mathrm{RBR},EA_{\mathrm{RBR}},BA_{\mathrm{RBR}},\rangle$$
where the elements are defined as follows:$Comp_{\mathrm{type}}\;\mathrm{RBR}=\langle Features_{\mathrm{RBR}},Flow_{\mathrm{spec}}\;\mathrm{RBR},Prop_{\mathrm{RBR}}\rangle$, with:
-$Features_{\mathrm{RBR}}$ =$IN_{p}$∪$OUT_{p}$, and $IN_{p}$, $OUT_{p}$∈ { event-data-ports},-$Flow_{\mathrm{spec}}\;\mathrm{RBR}$ = $\langle Flow_{p}\rangle$,-$Prop_{\mathrm{RBR}}$ = {$T_{e},AP\}.$
$Comp_{\mathrm{imp}}\;\mathrm{RBR}=\langle SC_{\mathrm{DataRBR}}\rangle$$EA_{\mathrm{RBR}}$ ={$Err_{p},E_{s},\to_{e},E_{e},Dur_{\mathrm{dist}},Occur_{\mathrm{dist}}$}$BA_{\mathrm{RBR}}$= {$B_{s},\to_{b}$}.

On the other hand, the DSS in our CAMI architecture is a CC, with multiple subcomponents and hence it is defined by its type, implementation and EA (no BA) as shown in Listing 2. Formally, it can be represented as follows:(5)$$DSS_{\mathrm{AADL}}=\langle Comp_{\mathrm{type}}\;\mathrm{DSS},Comp_{\mathrm{imp}}\;\mathrm{DSS},EA_{\mathrm{DSS}}\rangle$$
where the elements are defined as follows:$Comp_{\mathrm{type}}\;\mathrm{DSS}$ ={$Features_{\mathrm{DSS}}$, $Flow_{\mathrm{spec}}\;\mathrm{DSS}$, $Prop_{\mathrm{DSS}}$}, where:
-$Features_{\mathrm{DSS}}$ =$IN_{p}$∪$OUT_{p}$, and $IN_{p}$, $OUT_{p}$∈ {event-data-ports},-$Flow_{\mathrm{spec}}\;\mathrm{DSS}$ = $\langle Flow_{p}\rangle$,-$Prop_{\mathrm{DSS}}$ = {AP}.
$Comp_{\mathrm{imp}}\;\mathrm{DSS}$={$SC_{\mathrm{DSS}}$, $Pt_{\mathrm{DSS}}$, $Con_{\mathrm{DSS}}$, $Flow_{\mathrm{imp}}\;\mathrm{DSS}$}, where:
-$SC_{\mathrm{DSS}}$ = {CM,RBR,CBR},-$Pt_{\mathrm{DSS}}$ ={CM,RBR,CBR},-$Con_{\mathrm{DSS}}$ ={$IN_{\mathrm{pDSS}}\to IN_{\mathrm{pCM}}$, $OUT_{\mathrm{pCM}}\to IN_{\mathrm{pRBR}}$, $OUT_{\mathrm{pRBR}}\to IN_{\mathrm{pCBR}}$, $OUT_{\mathrm{pCBR}}\to OUT_{\mathrm{pDSS}}$},-$Flow_{\mathrm{imp}}\;\mathrm{DSS}$={$CM\to Flow_{p},RBR\to Flow_{p},CBR\to Flow_{p}$}.
$EA_{\mathrm{DSS}}=\left\{EM_{\mathrm{Comp}}\right\}$.

In the next sub-section, we present our semantic encoding of atomic and composite components, in terms of NSTA.

### 6.2. Formal Encoding of AADL Components as NSTA

Using the definition of AADL components given in Section 6.1, the formal definition of STA as $STA=\langle L,l_{0},A,V,C,E,I,\mu ,\gamma \rangle$, and of $NSTA={\left|\right|}_{i}STA_{i}$ (see Section 2.2), we define a semantic encoding of the AADL components, respectively, in terms of NSTA.

#### 6.2.1. Formal Encoding of AC

Any atomic component in AADL, defined by: $AC=\langle Comp_{\mathrm{typeAC}},Comp_{\mathrm{implAC}},EA_{\mathrm{AC}},BA_{\mathrm{AC}}\rangle$ is encoded as an NSTA as follows: $AC⇝NSTA_{AC}=AC_{\mathrm{iSTA}}\left|\right|AC_{\mathrm{aSTA}}$, where $AC_{\mathrm{iSTA}}$ is the so-called “Interface STA” of AC, which corresponds to $Comp_{\mathrm{typeAC}}$ and $Comp_{\mathrm{implAC}}$, whereas $AC_{\mathrm{aSTA}}$ is the “Behavioral STA” that encodes the EA and BA of an AC.

The **AC${}_{\mathrm{iSTA}}$** is defined according to a template STA (see Figure 8) with L∈{Idle,Op,Fail,start,stop}, $l_{0}=Idle$, Op corresponds to the Operational state of the RBR, start, stop represent the locations to initiate the synchronizations with $AC_{\mathrm{aSTA}}$ and E={Idle⟶start,start⟶Op,Op⟶stop,stop⟶Idle,Op⟶Fail,Fail⟶Idle}. This template is annotated with the following information:
-$V=out\_port\cup in\_port\cup \left\{PF,TF\right\}\cup SC_{\mathrm{Data}}\;$, where out_port and in_port represent the set of output and input ports ∈{data-ports, event-ports, event-data-ports}, respectively, and the Boolean variables, PF,TF, represent the error events associated with the transient failure and permanent failure of AC, plus the variable associated with $SC_{\mathrm{Data}}\in Comp\_imp$;-C={x} is the set of clocks that models the period and execution time of AC;  -A={start_ACi?,start_AC!,stop_AC!,stop_ACi!}∪{x=0}, where *A* is the set of synchronization channels associated with input-output ports ∈{event-data-ports, event-ports}, that is, channels start_AC!,stop_AC!, and the synchronization channels for the interface of the corresponding CC, that is, start_ACi?,stop_ACi! and the reset actions on *x*;-$E=\{Idle\overset{start\_ACi?\wedge x==T_{p}}{\to }start,start\overset{start\_AC!,x=0}{\to }Op,Op\overset{TF\_AC==1\vee PF\_AC==1}{\to }Fail,Op\overset{x==T_{e},stop\_AC!}{\to }stop,stop\overset{stop\_ACi!,x=0}{\to }Idle,Fail\overset{TF\_AC==0\wedge PF\_AC==0}{\to }Idle,Fail\overset{TF\_AC==1\wedge PF\_AC==1}{\to }Fail\}$, where *E* is defined by the template populated with A and guards that ensure the correctness of transitions.  -I(Idle)=$\left(x\leq T_{p}\right)$, if the dispatch protocol associated with AC is periodic, and $I\left(Op\right)=x\leq T_{e}$, where $T_{p}$ and $T_{e}$ represent the period and execution-time of AC;  -$P\left(Idle\right)=\mu_{1}$, and $P\left(Fail\right)=\mu_{2}$, where $P\left(Idle\right)=\mu_{1}$ represents the occurrence distribution of aperiodic event (if the dispatch protocol associated with AC is aperiodic), and $P\left(Fail\right)=\mu_{2}$ represents the probability of leaving location Fail;

The **AC${}_{\mathrm{aSTA}}$** is created in a similar way with:
-L={Wait,Op,TrF,PrF,Fail_ep,LReset,L1,L2},l0=Wait, where *L* comprises the set of states in EA and BA (Wait, Operational (Op), Transient Failure (TrF), Permanent Failure (PrF), Failed due to error propagation (Fail_ep), and reset location (LReset), plus additional committed locations (L1,L2) that ensure that receiving is deterministic in UPPAAL SMC;  -$A=\left\{start\_AC?,stop\_AC?\right\}\cup \{action_{\mathrm{BA},\mathrm{EA}}\left(\right),TF=0,TF\_AC=1,$PF_AC=1,reset_AC=0,reset_AC=1,err_pAC=0,err_pAC=1,err_p=1,y=0}, where *A* is composed of the actions defined in BA and EA (action ${}_{\mathrm{BA},\mathrm{EA}()}$), plus the synchronizations channels to concord with $AC_{\mathrm{iSTA}}$ (start_AC?,stop_AC?), and the reset of clock *y*;  -V={PF_AC,TF_AC,reset_AC,err_pAC}, where *V* consists of the set of error events defined in the EA, that is, PF_AC: Permanent Failure of AC, TF_AC: Transient Failure of AC, reset_AC: Reset of AC, err_pAC: error propagation of AC;  -C={y} is the clock that measures the time elapsed for reset action of a particular component;-$E=\{Wait\overset{start\_AC?}{\to }L1,L1\overset{TF\_AC=1,err\_pAC=1}{\to }TrF,L1\overset{PF\_AC=1,err\_pAC=1}{\to }PrF,L1\overset{}{\to }L2,L2\overset{}{\to }Op,Op\overset{stop\_AC?,action_{\mathrm{BA}}\left(\right)}{\to }Wait,TrF\overset{reset\_AC=1,y=0}{\to }LReset,TrF\overset{PF\_AC=1,err\_pAC=1,reset\_AC=0}{\to }PrF,LReset\overset{TF\_RBR=0,err\_pAC=0,reset\_AC=0}{\to }Wait,LReset\overset{PF\_AC=1,err\_pAC=1,reset\_AC=0}{\to }PrF,Wait\overset{err\_p==1}{\to }Fail\_ep\}$, where *E* consists of the transitions in EA, BA and those between L1 and L2;  -I(LReset)= $\left(y\leq Dur_{\mathrm{dist}(\mathrm{Reset})}\right)$;  -P(Wait)=μ, that is the occurrence-distribution of Wait;-$L1\overset{\gamma_{1}}{\to }L2$, $L1\overset{\gamma_{2}}{\to }TrF$, $L1\overset{\gamma_{3}}{\to }PrF,$ where $\gamma_{1},\gamma_{2},\gamma_{3}$, are defined according to the occurrence-distribution of the error events.□

#### 6.2.2. Formal Encoding of CC

The formal encoding of a CC defined by the tuple: $CC=\langle Comp_{\mathrm{typeCC}},Comp_{\mathrm{implCC}},EA_{\mathrm{CC}}\rangle$ is also a network of two synchronized STA, $CC_{\mathrm{NSTA}}=CC_{\mathrm{iSTA}}\left|\right|CC_{\mathrm{aSTA}}$, where $CC_{\mathrm{iSTA}}$ is the “interface” STA of the CC component, and $CC_{\mathrm{aSTA}}$ is the “annex” STA that encodes the information from the error annex in AADL.

The **CC${}_{\mathrm{iSTA}}$** is defined by formally encoding $\left(Comp_{\mathrm{typeCC}},Comp_{\mathrm{implCC}}\right)$, as follows:
-$L=\left\{Wait,Fail\right\}\overset{n}{\underset{i=1}{⋃}}\left\{L_{i}Sync\right\}\overset{n}{\underset{i=1}{⋃}}\left\{SC_{i}\right\}$, where L contains one location for each sub-component defined by SC, one additional location for each sub-component that ensures the correct synchronization, location Fail to model the component failure, and Wait to model the initial location;  -*E* is defined according to Con. For each connection in Con, we define two edges, $l⟶L_{i}Sync$ and $L_{i}Sync⟶l^{\prime }$, where l, l’∈L are locations created based on the sub-components for which the connections are defined, and $L_{i}Sync\in L$ is a location created for synchronization;  -$V=out\_port\cup in\_port\cup \left\{PF,TF\right\}\cup SC_{\mathrm{Data}}\;$, where out_port and in_port represent the set of output and input port variables ∈{data-ports, event-ports, event-data-ports}, respectively, and the Boolean variables, PF,TF, represent the error events associated with the transient failure and permanent failure of CC, plus the variable associated with $SC_{\mathrm{Data}}\in Comp\_imp$;  -C={x} if $T_{p}\neq \emptyset$;-*A* is defined based on the updates defined by MSM, the updates defined by $Flow_{\mathrm{imp}}$, the synchronizations defined by Con, the synchronization with $CC_{\mathrm{aSTA}}$, $AC_{\mathrm{aSTA}}$, and in case *C* is not void, we add the clock reset of the clock(s) in C;  -I(Wait)=$\left(x\leq T_{p}\right)$ if $T_{p}\neq \emptyset$;-P(l)=μ, where l∈L and μ is defined by Prop.

**CC${}_{\mathrm{aSTA}}$** is defined as follows:
-$L=E_{s}\in EA$, $l_{0}=s_{0e}\in E_{s}$, where $E_{s}$ is the set of states of EA;-$E=\to_{e}$;-A={TF_CC=1,TF_CC=0,PF_CC=1};-*V* is represented by the global variables defined in $CC_{\mathrm{iSTA}}$;-C=∅;-P(l)=μ, where l∈L and μ is defined by $Occur_{\mathrm{dist}}\in E_{\mathrm{prop}}$.All the other CC elements are transformed based on the encoding EA of AC.

Next, we show the rules instantiated on our previously selected AADL components of CAMI, that is, RBR and DSS, as examples of transforming AC and CC into corresponding STA. There are also some additional transitions defined which are not the direct result of applying the rules, but are needed due to the requirements of our modeling tool, UPPAAL SMC.

The $RBR_{\mathrm{AADL}}$ defined by Equation (Equation 4), is mapped into an NSTA (RBR${}_{\mathrm{NSTA}}$) as follows: $RBR_{\mathrm{NSTA}}$=$RBR_{\mathrm{iSTA}}\left|\right|RBR_{\mathrm{aSTA}}$ (Figure 9), where $RBR_{\mathrm{iSTA}}$ is the so-called “Interface STA” of RBR which corresponds to $Comp_{\mathrm{type}}\;\mathrm{RBR}$ and $Comp_{\mathrm{impl}}\;\mathrm{RBR}$, whereas $RBR_{\mathrm{aSTA}}$ is the “Annex STA” of RBR that encodes its EA and BA.

The $RBR_{\mathrm{iSTA}}$ is formally represented as a tuple, where:
-L={Idle,Start,Op,Fail}, l0={Idle}-A={start_RBRi?,start_RBR!,stop_RBR?}∪{x=1}-V={out_port,in_port,PF_RBR,TF_RBR}-C={x}-$E=\{Idle\overset{start\_RBRi?}{\to }start,start\overset{start\_RBR!,x=0}{\to }Op,Op\overset{TF\_RBR==1\vee PF\_RBR==1}{\to }Fail,Op\overset{x==1,stop\_RBR!}{\to }Idle,Fail\overset{TF\_RBR==0\wedge PF\_RBR==0}{\to }Idle,Fail\overset{TF\_RBR==1\wedge PF\_RBR==1}{\to }Fail\}$-I(Op)=(x≤1)  -P(Idle)=1, P(Fail)=1, given by γ

$RBR_{\mathrm{aSTA}}$ is defined in a similar way:
-L={Wait,Op,TrF,PrF,Fail_ep,LReset,L1,L2,LSync},{l0=Wait}-A={start_RBR?,stop_RBR?,stop_RBRi!}∪{rules(),TF_RBR={0,1},PF_RBR={1}, reset_RBR={0,1,}, err_pRBR={0,1}, err_p={1}, *y*=0}-$V=\{PF\_RBR,TF\_RBR,reset\_RBR,err\_pRBR,err_{p}\}$-C={y}-$E=\{Wait\overset{start\_RBR?}{\to }L1,L1\overset{TF\_RBR=1,err\_pRBR=1}{\to }TrF,L1\overset{PF\_RBR=1,err\_pRBR=1}{\to }PrF,L1\overset{}{\to }L2,L2\overset{}{\to }Op,Op\overset{stop\_RBR?,rules()}{\to }Lsync,Lsync\overset{stop\_RBRi!}{\to }Wait,TrF\overset{reset\_RBR=1,y=0}{\to }LReset,TrF\overset{PF\_RBR=1,err\_pRBR=1,reset\_RBR=0}{\to }PrF,LReset\overset{TF\_RBR=0,err\_pRBR=0,reset\_RBR=0}{\to }Wait,LReset\overset{PF\_RBR=1,err\_pRBR=1,reset\_RBR=0}{\to }PrF,Wait\overset{err\_p==1}{\to }Fail\_ep\}$-I(LReset)=y≤2-P(Wait)=10, given by μ-$L1\overset{0.9998}{\to }L2$, $L1\overset{0.001}{\to }TrF$, $L1\overset{0.001}{\to }PrF,$ assigned by γ

Similarly, the $DSS_{\mathrm{AADL}}$, shown in Listing 2, and represented by Equation (Equation 5), is mapped into an NSTA: $DSS_{\mathrm{AADL}}$ ⇝ $DSS_{\mathrm{NSTA}}$=$DSS_{\mathrm{iSTA}}\left|\right|DSS_{\mathrm{aSTA}}$ (Figure 10), where $DSS_{\mathrm{iSTA}}$ is the so-called “Interface STA” of DSS, which corresponds to $Comp_{\mathrm{type}}\;\mathrm{DSS}$ and $Comp_{\mathrm{impl}}\;\mathrm{DSS}$, whereas $DSS_{\mathrm{aSTA}}$ is the “Annex STA” that encodes the EA of CC.

The tuple elements of $DSS_{\mathrm{iSTA}}$ are as follows:
-L={Wait,CM,RBR,CBR,Fail,L1Sync,L2Sync,L3Sync,L4Sync}, l0={Wait}-A={start_DSSLC,start_CMi!,stop_CMi?,start_RBRi!,stop_RBRi?,start_CBRi!,stop_CBRi?,stop_DSSLC!,start_DSSCC!}∪{iCM_in=iDSSLC_in,iRBR_in=iCM_out,iCBR_in=iRBR_out,iDSSLC_out=iCBR_out,iDSSCC_in=iDSSLC_out}-V={iDSSLC_in,iCM_in,iRBR_in,iCBR_in,iDSSCC_in,iDSSLC_out,iCM_out,iRBR_out,iCBR_out,iDSSLC_out,PF_DSS,TF_DSS}-$E=\{Wait\overset{start\_DSSLC?}{\to }L1Sync,L1Sync\overset{start\_CMi!,iCM\_in=iDSSLC\_in}{\to }CM,CM\overset{stop\_CMi?}{\to }L2Sync,L2Sync\overset{start\_RBRi!,iRBR\_in=iCM\_out}{\to }RBR,RBR\overset{stop\_RBRi?}{\to }L3Sync,L3Sync\overset{start\_CBRi!,iCBR\_in=iRBR\_out}{\to }CBR,CBR\overset{stop\_CBRi?}{\to }L4Sync,L4Sync\overset{stop\_DSSLC!,iDSSLC\_out=iCBR\_out,iDSSCC\_in=iDSSLC\_out}{\to }Wait,CM\overset{(TF\_DSS=1\vee PF\_DSS=1),start\_DSSCC!}{\to }Fail,RBR\overset{(TF\_DSS=1\vee PF\_DSS=1),start\_DSSCC!}{\to }Fail,CBR\overset{(TF\_DSS=1\vee PF\_DSS=1),start\_DSSCC!}{\to }Fail,Fail\overset{\left(TF\_DSS==1\vee PF\_DSS==1\right)}{\to }Fail,Fail\overset{\left(TF\_DSS==0\wedge PF\_DSS==0\right)}{\to }Wait\}$-P(Wait)=10, P(CM)=10, P(RBR)=10, P(CBR)=10, P(Fail)=1
$EA_{\mathrm{CC}}⇝$
$DSS_{\mathrm{aSTA}}$


$DSS_{\mathrm{aSTA}}$ has the following syntactic elements:-L={Wait,TrF,PrF}, l0={Wait}-
A={TF_DSS={0,1},PF_DSS={1}}
-
V={TF_DSS,TF_CM,TF_RBR,TF_CBR,PF_CM,PF_RBR,PF_CBR,PF_DSS}
-
$E=\{Wait\overset{TF\_CM==1\wedge TF\_RBR==1\wedge TF\_CBR==1,TF\_DSS=1}{\to }TrF,Wait\overset{PF\_CM==1\wedge PF\_RBR==1\wedge PF\_CBR==1,PF\_DSS=1}{\to }PrF,PrF\overset{PF\_DSS==1}{\to }PrF,TrF\overset{TF\_CM==0\vee TF\_RBR==0\vee TF\_CBR==0,TF\_DSS=0}{\to }Wait\}$
-
P(Wait)=10,P(TrF)=10,P(PrF)=10


It should be noted that in the CAMI architecture, the semantic encoding of its components are restricted to the scope of the verification, and hence the components like the Database, UI, Security, and Privacy are not encoded as STA. The semantic encoding produces a complex NSTA comprising 32 STA, out of which 18 STA are produced by encoding the 10 AC of CAMI (four sensors: one for detecting pulse data deviation, two for fall detection and one for fire detection, data collector, Message Queue, RBR, CBR, daily activity detection, fuzzy logic) and the remaining 12 by encoding six CC (Local Processor, Cloud Processor, DSS (Local and Cloud), and Context modeling in DSS( Local and Cloud) of the AADL model of CAMI. On the other hand, the NSTA model of the minimum architecture configuration is comprised of only 18 STAs and is shown to be scalable with exhaustive analysis.

## 7. AAL Architecture Verification and Discussion

In this section, we verify if the minimum configuration architecture, and the most complex one, the CAMI architecture introduced in Section 4, satisfy their requirements as described in the same section, respectively. We apply exhaustive model checking for the first case and statistical model checking in the second case.

### 7.1. Exhaustive Verification of the Minimum Configuration Using UPPAAL

The results of the exhaustive verification of the minimum configuration architecture using the UPPAAL model checker are tabulated in Table 1. To check that our system meets its requirements, we employ a monitor STA that monitors the sensor values, the respective DSS output, and the corresponding clock. The monitor automaton for $R1_{\mathrm{Arch}1}$ is shown in Figure 11. As described, we start the monitoring clock *s1* when the pulse sensor produces the data, marked by the transition to *L2* triggered by the synchronization channel, and we stop the clock when a decision is produced by the cloud DSS. Similar monitors have been employed for $R2_{\mathrm{Arch}1}$.

We have used queries of the form *A leads to B* for our analysis and therefore a pre-check of each corresponding “A”, being reachable is first carried out. Moreover, since our model is an STA model where each component has associated failure probabilities and failure of a component does not yield the intended results during exhaustive verification, we verify the properties considering all the components are operational. $R1_{\mathrm{Arch}1}$ requires that if the pulse is high and the user is not exercising, then an abnormal pulse alert is raised to the caregiver within 20 s. In $R2_{\mathrm{Arch}1}$, we verify that if the fall sensor detects a fall event, then a fall alert is raised to the caregiver within 20 s. The aforementioned requirements are safety requirements of the system and it is shown that these requirements are met provided all the system components are operational. However, its equally important to mention that if the real-time constraints are relaxed, for instance, if we check if these requirements are met within 10 s, the model-checker obviously returns a fail for the corresponding query (shown in Table 2).

### 7.2. Statistical Verification of the CAMI architecture Using UPPAAL SMC

In the case of CAMI architecture, which is the most complex instantiation of our proposed generic architecture, exhaustive verification does not scale, and hence we chose to verify the CAMI system requirements using UPPAAL SMC [7], the statistical extension of UPPAAL model checker to perform probabilistic analysis. To verify the functional requirements, we employ monitor STA to monitor the sensor values, the respective DSS output and the corresponding clock. For instance, an example of monitor STA for $R1_{\mathrm{CAMI}}$ is given in Figure 12. As shown, we start the monitoring clock s1 when the fire sensor produces the data, marked by transition to L2 triggered by the synchronization channel and we stop the clock when a decision is produced by local DSS or the cloud DSS. Similar monitors are employed for $R2_{\mathrm{CAMI}}$, $R3_{\mathrm{CAMI}}$, $R4_{\mathrm{CAMI}}$, and $R5_{\mathrm{CAMI}}$.

The verification results are tabulated in Table 3. The CAMI architecture model satisfies all the requirements with probabilities close to 1 with a high confidence within four minutes until a result is returned. As in the other case, since most queries contain terms of the form *A*imply*B*, we first check the reachability of A. From the analysis, it follows that the probability of the cloud DSS to get activated ($(R6_{\mathrm{CAMI}}$) is [0.01, 0.04]. This is justified by the fact that it becomes active only when the local DSS has failed and the failure probability of local DSS is between [0.01, 0.04] for a simulation over 1000 time units, which is a safe value to assume for safety-critical systems.

### 7.3. Comparison of the Proposed Approach With the Model-Checker PRISM

In this section, we show a small-scale modeling of the CAMI architecture using the model checker, PRISM [8] to compare the results that we have obtained with UPPAAL SMC. For details regarding the transformation of AADL to PRISM, please refer our previous work [30]. To ensure scalability of the analysis, we model only the scenarios of fire and fall where the rule-based reasoning takes the action of forwarding the respective alerts to firefighter and caregivers, respectively (R1${}_{\mathrm{CAMI}}$, R2${}_{\mathrm{CAMI}}$ and R3${}_{\mathrm{CAMI}}$).

In Listing 3, we show an excerpt of the RBR module modeled as a Probabilistic Timed Automaton (PTA) in PRISM model checker. In line 3, we define the variable *s* representing the state of the system: s = 0 (*Idle*), s = 1 (*Op*), s = 3 (*TrF*), s = 4 (*PrF*), and s = 5 (*Fail_ep*). In the following lines, we define the event variables for transient failure (TF_RBR), permanent failure (PF_RBR), failure due to error propagation (Fail_ep_RBR) and the reset event (reset_RBR). We also define a clock variable *x* to model the RBR’s execution time ( an invariant associated with state *Op*). In lines 13–22, we model the functional and error behavior of the component as discussed in the previous sections. We show the cases of a fire event and fall event generated separately, where subsequent alerts to firefighter and caregiver informing of the respective events are sent. We also illustrate the case where fire and fall events occur together, where both the events need to be sent to both caregiver and firefighter.

Listing 3: An excerpt of the PRISM model of an RBR

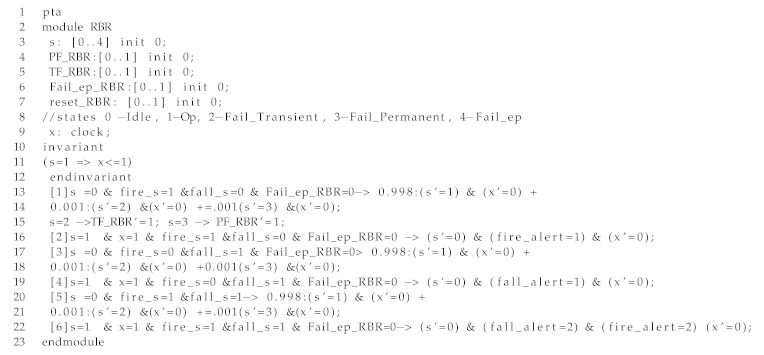



The analysis results are presented in Table 4. The CAMI requirements are formulated as Probabilistic Computation Tree Logic (PCTL) queries. Moreover, since the PRISM model checker returns the result for the initial state of the model by default, we employ *filters* to verify the properties over all states. R1${}_{\mathrm{CAMI}}$ ensures that if a fall event is raised by fall sensor, then the fall alert is communicated to the caregiver within 20 time units, provided that none of the components has failed. Similarly, in R2${}_{\mathrm{CAMI}}$, we show the case of a fire event being communicated to a firefighter within 20 time units. In R3${}_{\mathrm{CAMI}}$, we show the case of fire and fall events simultaneously raised, where we need to communicate both events to firefighter and caregiver within a real-time deadline of 20 time units.

We can start the comparison of UPPAAL, PRISM, and UPPAAL SMC verification with the following known facts: (a) PRISM allows exhaustive model-checking of probabilistic systems (b) For UPPAAL, although the results are exhaustive, the probabilistic view of the system is discarded and (c) UPPAAL SMC allows us to analyze probabilistic systems, however the analysis is simulation-based, that is, the guarantees are obtained by simulating the system for a finite number of runs and hence not exhaustive or fully guaranteed. In our case, the major disadvantage we found with PRISM was that it did not scale well and hence we had to limit the complexity of our CAMI architecture for analysis purposes. Moreover, it lacks a graphical GUI, as compared to UPPAAL systems and do not provide simulation-based analysis for PTA models, thereby limiting the diagnosis of the models.

## 8. Discussion

The approach presented in this paper paves the way for the development of formally assured future intelligent AAL solutions that integrate multiple functionalities. Our approach can be applied at earlier design stages to capture potential errors that can propagate across the development stages, which may result in significant re-engineering costs. Our architecture description framework (AADL) has a commercially available tool support, OSATE [31] for automated modeling, and provides some preliminary architecture-level analysis. It also allows us to model the behavior of the architecture components via a behavior annex and encode the probabilities of failure of various components, via the error annex. However, AADL has its limitations of expressing complex behaviors of algorithms such as CBR, which we have omitted in this work.

There are two analysis approaches presented in this paper: (1) involving exhaustive model checking, with the UPPAAL tool (2) involving statistical model checking with UPPAAL SMC, and (3) involving probabilistic model checking with PRISM. The analysis approaches are chosen based on the system complexity. If the architecture model is scalable with exhaustive model checking, then the latter can be applied. Although the exhaustive verification results obtained by UPPAAL are accurate, one cannot take into account the probabilistic behavior of our systems. In comparison, PRISM handles probabilistic systems and carries out an exhaustive analysis, however its scalability is considerably reduced if compared to UPPAAL SMC. In the case of complex models that need to be analyzed for stochastic behaviors, the user can opt for simulation-based approaches, although it does not yield a 100% accuracy. The verification results shown in this paper are specific to our architecture models, however one can use the approach to verify any set of requirements for various architecture types created based on the generic architectural model defined in this work. In case of exhaustive model-checking, the results are derived assuming that all components are operational such that we devoid the system of its probabilistic failure behavior. For statistical model checking, it is worth mentioning that the results are derived assuming high reliability of individual architecture components and considering specific values for the periods and execution times. Nevertheless, taking into account the wide variety of available sensors and other components, we can easily adapt the values to account for the requirements of any specific architecture.

In addition, the approach presented in this paper is generic and easily extensible. Our modeling methodology based on AADL abstract components can be extended to suit particular run-time representations of the system. The AADL semantics as networks of STA is also generic and can be extended to accommodate other AADL properties that we have not accounted for in this work. We expect that similar results can be reproduced if the approach followed in this paper is used in other integrated AAL solutions.

## 9. Related Work

In recent years, there has been a lot of work in the area of AAL due to the need of supporting an increased elderly population [32]. Moreover, many functionalities that need to be tackled by AAL solutions are of a safety-critical nature, e.g., health emergencies like cardiac arrest, falls of the elderly, and home emergencies like fires at home, etc. [33], therefore work on their modeling and analysis is fully justified.

A study on existing AAL architectures shows that there are certain architecture types that address the construction of integrative AAL applications, some of the common ones being: Multi-Agent Systems (MAS) [34,35,36], and Cloud-based [37,38] and Internet-of-Things (IoT) centric solutions [39].

**Agent-based architectures:** These are the most commonly used architectures for AAL applications, based of their flexibility, autonomy, adaptability, better response, and service continuity due to their distributed nature. Some examples of health-care frameworks that rely on a distributed agent architecture are described in the literature [34,40]. However, the agent-based architectures have some drawbacks: (i) restricted communication protocols for agent communication and the delay overhead in taking a collective decision; and (ii) maintaining the consistency.**Cloud-based AAL solutions:** This category includes AAL solutions that leverage the potential of cloud computing for context modeling, intelligent decision making, and data-storage usage. Our architecture follows the design paradigms of Cloud-based AAL solutions, where the cloud is utilized for intelligent, context-aware decision making, also as a data store, and it is also augmented with local processing schemes to guarantee real-time properties. In many situations, cloud services cannot guarantee hard-real time properties, the cloud being a backup that gets activated only when the primary one has failed.

The formal assurance of AAL systems has been the focus of some related research in the recent years. Parente et al. provide a list of various formal methods that can be used for AAL systems [41]. In another interesting work, Rodrigues et al. [4] performed a dependability analysis of AAL architectures using UML and PRISM. Other interesting research work uses temporal reasoning [3,42] and Markov Decision Processes to formally verify the reliability of AAL systems [43]. Although these approaches target the formal analysis of AAL systems, most of the above work addresses only simple scenarios, and are not used to analyze complex behaviors resulting from integrating critical AAL functions (e.g., fire and fall), as well as their decision making. In addition, these approaches do not aim to develop an overall framework for the verification of AAL systems, starting from an integrated architectural design followed by a verification strategy, as proposed in this paper.

The use of Architecture Description Languages (ADL) to specify AAL designs has not been exercised previously, yet such languages are commonly employed in the design of automotive or automation systems. There have also been approaches to formally verify AADL designs in other domains. The transformation approach from AADL to TA or variants has been already addressed by related work [44,45,46]. Although these approaches rely on automated verification techniques, there is a lack of focus on abstract components/patterns with stochastic properties. In addition, these approaches also suffer from state-space explosion, therefore they might not scale well to complex AAL designs. Nevertheless, there is interesting research that deals with stochastic properties and statistical model checking for the analysis of extended AADL models. One such example is in the work of Bruintjes et al. [47], where the authors have used the SMC approach for timed reachability analysis of extended AADL designs. Although our approach also focuses on linear systems, it is different from the mentioned work in the fact that we focus on abstract components, and also introduce BA modeling for capturing the functional behavior of our modules, specifically for modeling the behavior of intelligent DSS. In their work, Bruintjes et al. used the SLIM Language, which is strongly based on AADL and is specific to avionics and the automotive industry, including the error behavior and modes. However, we use the AADL core language with its standardized annex sets (EA and BA) for the architecture specification, which enables us to represent the functional and error behaviors, together with architecture model. The abstract component-based modeling also brings extensiblity and reusability to our approach. Moreover, the authors only consider the event occurrences or delay variations using uniform or exponential distributions, whereas by employing our user-defined properties, we can also specify other distributions. Furthermore, the approach of Bruintjes et al. only deals with evaluation of time-bounded queries, whereas we also evaluate properties like reliability, data consistency, etc., along with timeliness. Another interesting work [48], possibly carried out in parallel with our work, employed statistical model checking using UPPAAL SMC to evaluate the performance of nonlinear hybrid models with uncertainty, modeled in extended AADL. Although the approach is not specific to the AAL domain, it is promising with respect to specifying complex cyber physical systems considering the uncertainties of the physical environment. Unlike our model, the authors used Priced Timed Automata (PTA) models. In comparison, our approach considers only linear models that evolve continuously (yet the analysis is carried out in discrete time due to sampling of continuous data). In brief, the two approaches resemble each other, yet our approach is contained in the core language of AADL (as different from the mentioned work where the authors resort to other annexes integrated in OSATE), is tailored to systems that contain AI components, and assumes the random failure of various components, which is not considered in the related work.

## 10. Conclusions and Future Work

In this paper, we have proposed a generic AAL architecture and its intelligent Decision Support System that can tackle a multitude of functionalities by analyzing the interdependencies between simultaneously occurring events. We have also presented three specific instantiantions of the generic model, following an increasing order of complexity. In addition, we have also presented a framework for modeling and verification of our specific integrated AAL system architectures. To provide formal analysis for the AAL systems, we have semantically encoded the AADL model as networks of stochastic timed automata.

We show that the resulting formal models are analyzable exhaustively with UPPAAL/PRISM or statistically with UPPAAL SMC (chosen based on system complexity), to ensure that the required functional behavior is met. Our contribution is generic and paves the way for the development of formally-assured intelligent AAL system architectures.

The framework is intended to augment existing AAL solutions with formal analysis support and provide analysis prior to implementation. Such an analysis is crucial in domains such as AAL, which are real-time, safety-critical ones, and require high levels of dependability. Due to the heterogeneity of components available in the AAL domain, the component failure probabilities, periods and execution times are not chosen with respect to to any specific components, nevertheless the results presented in the paper are promising because the abstract components that have been proposed can be refined further.

In the future, we plan to enhance our DSS model with more rules for RBR and full functionality support of CBR and activity recognition, thereby providing an extensive analysis of AAL systems behaviors in possible critical scenarios. Another interesting direction to proceed with is providing automated tool support for the semantic mapping. We are also currently investigating formal modeling and analysis of distributed versions of the integrated architectures for AAL.

## Figures and Tables

**Figure 1 sensors-19-05057-f001:**
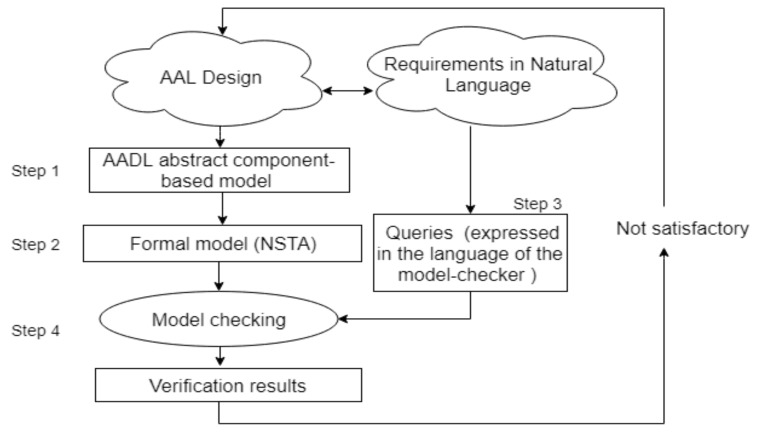
Methodology overview. Ambient Assisted Living (AAL), Architecture Analysis and Design language (AADL), stochastic timed automaton (STA), network of interacting STA (NSTA).

**Figure 2 sensors-19-05057-f002:**
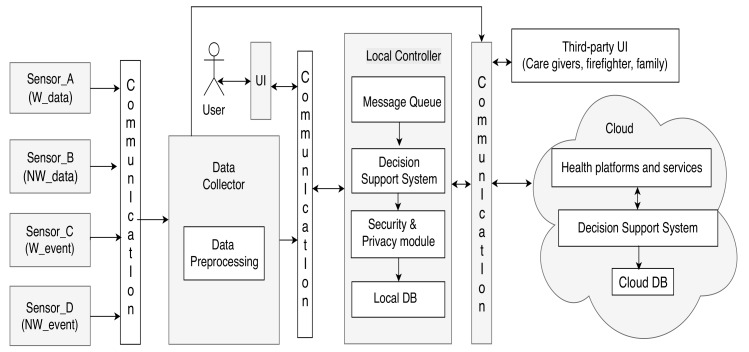
The generic Ambient Assisted Living system architecture.

**Figure 3 sensors-19-05057-f003:**
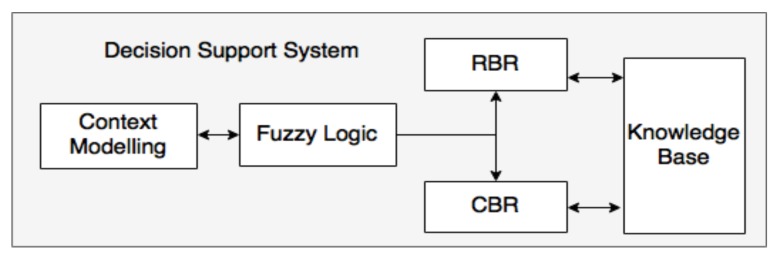
The decision support system (DSS) architecture. Rule-based reasoning (RBR) and case-based reasoning (CBR).

**Figure 4 sensors-19-05057-f004:**
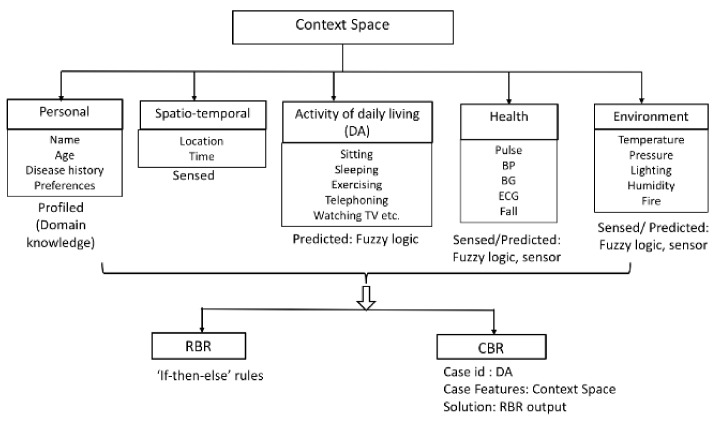
Internals of the DSS architecture (List of Artificial Intelligent techniques).

**Figure 5 sensors-19-05057-f005:**
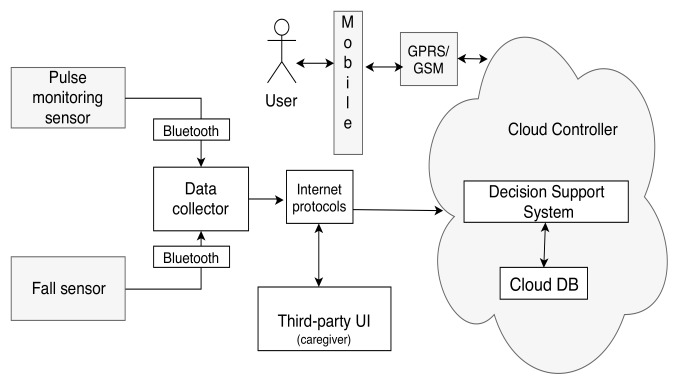
Category 1: A minimal configuration.

**Figure 6 sensors-19-05057-f006:**
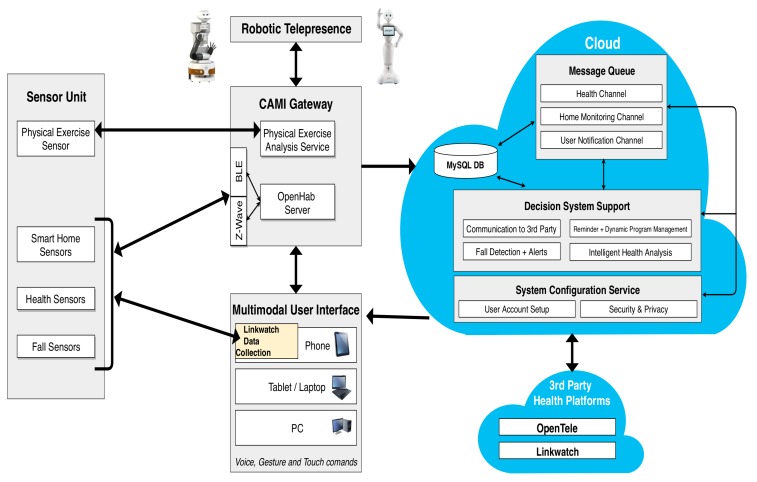
Category 2: An intermediate configuration.

**Figure 7 sensors-19-05057-f007:**
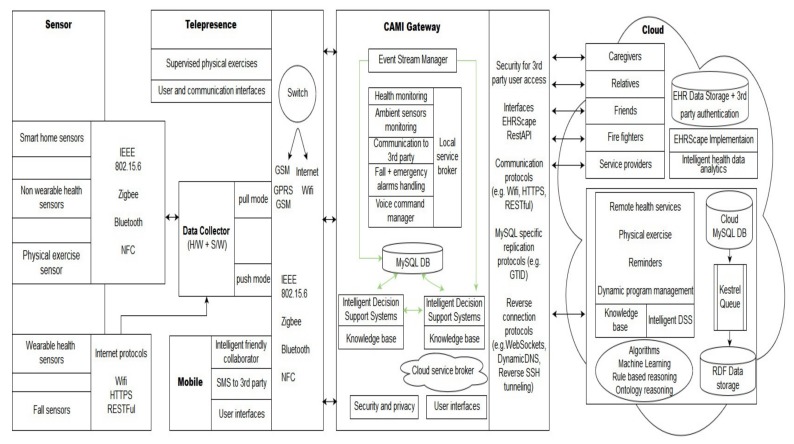
Category 3: A complex configuration: The CAMI AAL System Architecture [2].

**Figure 8 sensors-19-05057-f008:**
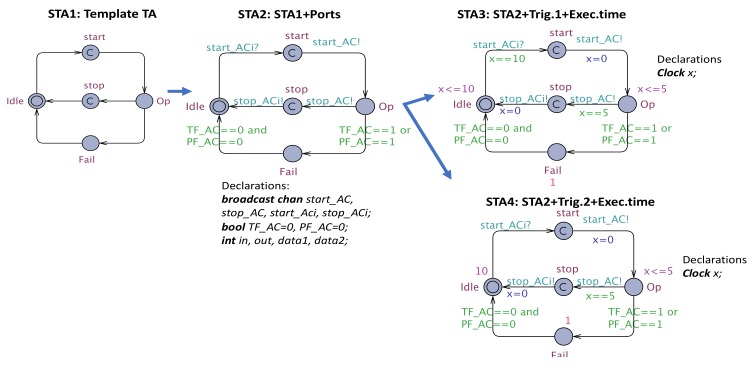
Step-by-step formulation of $AC_{\mathrm{iSTA}}$.

**Figure 9 sensors-19-05057-f009:**
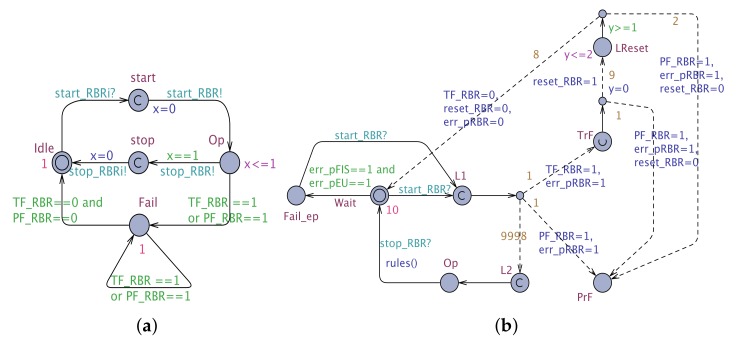
The STA for the RBR. (**a**) Interface STA ($RBR_{\mathrm{iSTA}}$); (**b**) Annex STA ($RBR_{\mathrm{aSTA}}$).

**Figure 10 sensors-19-05057-f010:**
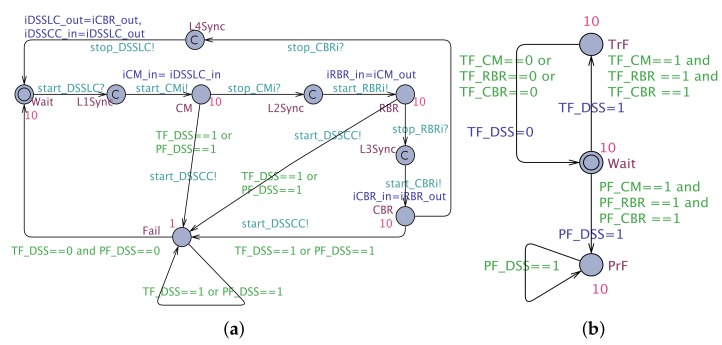
The STA for the DSS. (**a**) Interface STA ($DSS_{\mathrm{iSTA}}$); (**b**) Annex STA ($DSS_{\mathrm{aSTA}})$.

**Figure 11 sensors-19-05057-f011:**
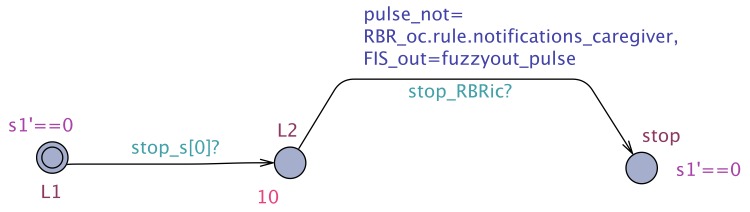
The monitor automaton for requirement $R1_{\mathrm{Arch}1}$.

**Figure 12 sensors-19-05057-f012:**
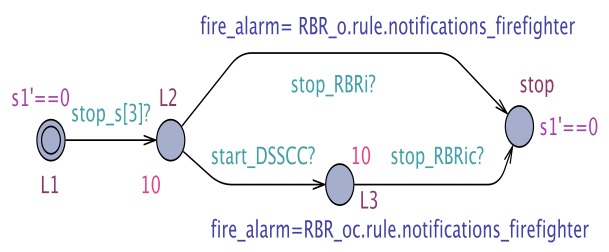
The monitor automaton for requirement $R1_{\mathrm{CAMI}}$.

**Table 1 sensors-19-05057-t001:** UPPAAL analysis results for the minimum configuration architecture.

REquation	Query	Result
R1${}_{\mathrm{Arch}1}$	(110≤sd_w.data_val≤300andADL=1and M_pulse.FIS_out==3andop_DC==1 andop_fuzzy==1andop_RBR==1) →M_pulse.pulse_not==3andM_pulse.s1≤20	Pass
	E<>(110≤sd_w.data_val≤300 and and ADL=1 M_pulse.FIS_out==3 and op_DC==1 and op_fuzzy==1 and op_RBR==1)	Pass
R2${}_{\mathrm{Arch}1}$	(se_w.fall==1andop_DC==1 andop_EU==1andop_RBR==1) →M_fall.fall_not==7andM_fall.s1≤20	Pass
	E<>(se_w.fall==1 and op_DC==1 and op_EU==1 and op_RBR==1)	Pass

**Table 2 sensors-19-05057-t002:** UPPAAL analysis: A case where real-time constraints are not met.

REquation	Query	Result
R1${}_{\mathrm{Arch}1}$	(110≤sd_w.data_val≤300andADL=1and M_pulse.FIS_out==3andop_DC==1 andop_fuzzy==1andop_RBR==1) →M_pulse.pulse_not==3andM_pulse.s1≤10	Fail

**Table 3 sensors-19-05057-t003:** UPPAAL SMC analysis results of CAMI.

REquation	Query	Result	Runs
R1${}_{\mathrm{CAMI}}$	Pr[<=1000]([]((M_fire.fire_alarm==1) imply(se_nw.fire==1 and M_fire.s1<=20)))	Pr [0.99975,1] confidence 0.998	3868
	Pr[<=1000](<>(M_fire.fire_alarm==1))	Pr [0.99975,1] confidence 0.998	4901
R2${}_{\mathrm{CAMI}}$	Pr[<=1000]([]((M_fall.fall_not==7) imply((se_w.fall==1 or sd_nw.data_val==1) and(M_fall.s1<=20))))	Pr [0.99975,1] confidence 0.998	3868
	Pr[<=1000](<>(M_fire.fire_alarm==1))	Pr [0.99975,1] confidence 0.998	4901
R3${}_{\mathrm{CAMI}}$	Pr[<=1000]([](M_firefall.fire_not==2and M_firefall.fall_not==2imply ((se_w.fall==1 or sd_nw.data_val==1) and se_nw.fire==1 and M_firefall.s1<=20))	Pr [0.99975,1] confidence 0.998	3868
	Pr[<=1000](<>(Pr[<=100](<>(M_firefall. fall_not==2 and M_firefall.fire_not==2))	Pr [0.99975,1] confidence 0.998	7905
R4${}_{\mathrm{CAMI}}$	Pr[<=1000]([]((M_pulse.pulse_not==3) imply(110<=sd_w.data_val<=300and M_pulse.FIS_out==3andADL==1and upro.disease_history==3 and M_pulse.s1<=20))	Pr [0.99975,1] confidence 0.998	3868
	Pr[<=1000](<>(M_pulse.pulse_not==3))	Pr [0.99975,1] confidence 0.998	3868
R5${}_{\mathrm{CAMI}}$	Pr[<=1000]([](M_consistency.stop imply $\left(RBR\_om==iCBRCC_{m}))\right)$	Pr [0.99975,1] confidence 0.998	3868
	Pr[<=1000](<>(M_consistency.stop))	Pr [0.99975,1] confidence 0.998	5777
R6${}_{\mathrm{CAMI}}$	Pr[<=1000]([](INT_CC.DSSCC imply PF_DSS==1))	Pr [0.99975,1] confidence 0.998	3868
	Pr[<=1000](<>(INT_CC.DSSCC))	Pr [0.01,0.04] confidence 0.998	2885

**Table 4 sensors-19-05057-t004:** PRISM verification results.

REquation	Query	Result	Time (s)
R1${}_{\mathrm{CAMI}}$	[l] filter(forall,fall_s=1&fire_s=0&PF_RBR=0 →P≥0.999 [F((fall_alert=1)&(x≤20)&(fall_fail=0)&(DC_fail=0)]	satisfied	2001.7
R2${}_{\mathrm{CAMI}}$	[l] filter(forall,fall_s=0&fire_s=1&PF_RBR=0 →P≥0.999 [F((fire_alert=1)&(x≤20)&(fire_fail=0)&(DC_fail=0)]	satisfied	2001.7
R3${}_{\mathrm{CAMI}}$	[l] filter(forall,fall_s=1&fire_s=1&PF_RBR=0 →P≥0.999 [F((fire_fall_alert=2)&(x≤20)&(fire_fail=0)&(fall_fail=0) &(DC_fail=0)]	satisfied	3500.13

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
