# Peer review of "A Model-Checking-Based Framework for Analyzing Ambient Assisted Living Solutions"

_sensors, 2019, doi:10.3390/s19225057_

Round 1

Reviewer 1 Report

In "A Model-Checking-Based Framework For Analyzing Ambient Assisted Living Solutions" the authors present, with a too-long paper and with too many formal details, a framework to model, verify and validate the design of AAL environment.

This goal was obtained describing the architecture of the system by an architecture description language (AADL) and analyzing the formal description with UPPAAL's tools.

They present 3 cases with an increasing level of complexity: 1. minimal configuration; 2. intermediate; 3. complex.

The premise of this work is that the proposed approach is important to detect possible errors related to the integration of a multitude of assisted-living functionalities and heterogenous AAL systems.

However, in the introduction, it is not very clear which errors can be detected and corrected.

Reading the example proposed, after a long formal discussion, it seems to me that the results obtained are very trivial - at least in case of study n.1 (row n. 912 to 920) - and not very interesting.

Furthermore, other approaches are not presented and any comparison is impossible.

There are not quantifiable results that it is possible to compare or evaluate.

It is not my intention to discuss the suitability of the proposed approach, that could be appropriate, but the purpose of the paper.

If the paper aims to present the work carried out by authors in the framework of the AAL Joint Programme (that is the more appropriate choice, in my opinion), it seems to me too much long and detailed. My suggestions are:
- explain much better the reason justifying the proposed approach, identifying situations in which the problems that the design of complex AAL systems can generate are much more evident
- shorten the paper, strongly limiting the use of code listing and formal descriptions that are related to the use of a specific tool, focusing on the advantages that can derive from the use of such techniques of description and analysis.

If it has more scientific ambitions, I think that the article must be reviewed in its overall form, clarifying the performances that can be achieved with such an approach and comparing them with other alternative solutions.

Author Response

First, we want to thank both reviewers for their time dedicated to our paper and for their valuable comments. We did our best to address them all, as we show below.

Review 1

In "A Model-Checking-Based Framework For Analyzing Ambient Assisted Living Solutions" the authors present, with a too-long paper and with too many formal details, a framework to model, verify and validate the design of AAL environment.

This goal was obtained describing the architecture of the system by an architecture description language (AADL) and analyzing the formal description with UPPAAL's tools.

They present 3 cases with an increasing level of complexity: 1. minimal configuration; 2. intermediate; 3. complex.

The premise of this work is that the proposed approach is important to detect possible errors related to the integration of a multitude of assisted-living functionalities and heterogenous AAL systems.

However, in the introduction, it is not very clear which errors can be detected and corrected.

Thank you for your comment. We have addressed this in the introduction by clearly specifying the errors that our model checking frameworks detect including functionality errors, timeliness errors, data inconsistencies and non-fault-tolerant behavior (Line 63). We have also addressed in results a case where our model checker detects a timed functional error (Table2, Page 22) .

Reading the example proposed, after a long formal discussion, it seems to me that the results obtained are very trivial - at least in case of study n.1 (row n. 912 to 920) - and not very interesting.

In rows 912-920, we are just elaborating the used formal notations used. The text does not relate to verification results.

Furthermore, other approaches are not presented and any comparison is impossible.

This is a valuable comment, thank you. On page 24, we have added a separate section (6.3) where we compare our results with those obtained if one applies exhaustive probabilistic model checking by employing the PRISM model checker. In addition, in the related work section, we have done an extensive comparison with other approaches. (page 26, lines 993-1004)

There are not quantifiable results that it is possible to compare or evaluate.

In connection also to the comment above, we have evaluated and compared our analysis results with those obtained from PRISM model checker, which is a different approach to verification, which employs exhaustive probabilistic model checking rather than statistical model checking.

It is not my intention to discuss the suitability of the proposed approach, that could be appropriate, but the purpose of the paper.

If the paper aims to present the work carried out by authors in the framework of the AAL Joint Programme (that is the more appropriate choice, in my opinion), it seems to me too much long and detailed. My suggestions are:
- explain much better the reason justifying the proposed approach, identifying situations in which the problems that the design of complex AAL systems can generate are much more evident

Thank you, we clarify this issue here. The complex framework presented in this paper is the architecture model of the CAMI project (give the home page address), an AAL joint program that funded our research. We have identified scenarios where the integration of complex functionalities are not trivial at all, and also referenced our previous paper ‘Do we need an integrated solutions for Ambient Assisted Living’ , published in UCaMI 2016 where we present in detail why and when the integration of critical functionalities are crucial in AAL systems. The formal analysis of many such scenarios are presented in this paper, which to our knowledge is one of the very few endeavors of formally analyzing AAL systems (if not the only one). Since the architecture itself has been a contribution in our CAMI project, and since it is general and customizable, we have shown different instantiations of this proposed architecture, with increasing complexity, to emphasize the following: (1) that one can “build” AAL systems easily, by selecting the desired functionality, and (2) to show how one can formally verify these various configurations, from simple ones on which exhaustive model checking scales, to more complex ones in which probabilities of failure are considered, and statistical model checking is the only option.  To achieve these results, we needed to first define the formal semantics of the AADL constructs in which our AAL architectures are specified. 

- shorten the paper, strongly limiting the use of code listing and formal descriptions that are related to the use of a specific tool, focusing on the advantages that can derive from the use of such techniques of description and analysis.

To address your concern, we are keeping in the paper only the required code listings to understand the Others (such as internals of DSS functioning and appendix of full AADL code of RBR ) that contain detailed information have been removed.

If it has more scientific ambitions, I think that the article must be reviewed in its overall form, clarifying the performances that can be achieved with such an approach and comparing them with other alternative solutions.

Indeed, the paper has scientific ambitious, please see the comments above. To address your comment, we compared our SMC analysis results with those obtained by employing exhaustive model checking with PRISM. This addition can be found in Section 6.3

Reviewer 2 Report

The paper details a model-checking based framework that facilitates the formal analysis of cyber-physical systems in the AAL domain. The paper is very well written (especially with regards to its first half, but the latter part should be revised in order to eliminate typos, punctuation errors and to rephrase some sentences; a few examples are found below). The approach appears scalable, which is very well argued by the authors using the selection of three proposed systems of different complexity. One area which feels not as well covered by the paper is the translation between the actual CPS and its formalization, but perhaps that is out of scope for the presented work. In the following, a few observations that were raised while perusing the paper:
- Section 1.2 could be integrated into 1.1 as it does not really present a contribution
- Line 202-203: double "the", line 207: acronym typo
- Most medical literature uses the 60-100 range for normal heart rate. Maybe you should explain why the 60-120 range was used?
- Text in Figures 6, 7 and 8 should be increased, as it will be difficult to read in print. Text size should probably be closer to that of the regular font used.
- Maybe create subsections for presenting Atomic and Composite Components
- The published version of the paper will have the line numbers removed, so have a separate numbering within the listing, if you wish to refer to them directly (e.g. lines 423, 425)
- Formal encoding of AC (line 655) and Formal encoding of CC (line 708) should have their own paragraphs/subsection numbers.
- Line 931 should probably read Table 2, line 949, automated is doubled, typos on line 951, 952, 955
- Double check paragraphs starting at line 978, 984.
- Line 1026 "CPS systems" (systems is redundant?)

Author Response

First, we want to thank both reviewers for their time dedicated to our paper and for their valuable comments. We did our best to address them all, as we show below.

The paper details a model-checking based framework that facilitates the formal analysis of cyber-physical systems in the AAL domain. The paper is very well written (especially with regards to its first half, but the latter part should be revised in order to eliminate typos, punctuation errors and to rephrase some sentences; a few examples are found below). The approach appears scalable, which is very well argued by the authors using the selection of three proposed systems of different complexity. One area which feels not as well covered by the paper is the translation between the actual CPS and its formalization, but perhaps that is out of scope for the presented work. In the following, a few observations that were raised while perusing the paper:
- Section 1.2 could be integrated into 1.1 as it does not really present a contribution

Thank you for the comment. We addressed it by integrating Section 1.2 into 1.1, as you suggested.

- Line 202-203: double "the", line 207: acronym typo

-The acronym typo has been removed.

- Most medical literature uses the 60-100 range for normal heart rate. Maybe you should explain why the 60-120 range was used?

-Indeed an important point. However, 120 was a typo, and we have fixed it by using the normal 60-100 range.

- Text in Figures 6, 7 and 8 should be increased, as it will be difficult to read in print. Text size should probably be closer to that of the regular font used.

- Thank you, the figures have been adjusted maximum to fit the space and size

- Maybe create subsections for presenting Atomic and Composite Components

- We have created separate subsections 4.1 and 4.2

- The published version of the paper will have the line numbers removed, so have a separate numbering within the listing, if you wish to refer to them directly (e.g. lines 423, 425)

We added new numbering for the listing

- Formal encoding of AC (line 655) and Formal encoding of CC (line 708) should have their own paragraphs/subsection numbers.

-We added separate subsections for AC and CC, that is, 5.2.1 and 5.2.2

- Line 931 should probably read Table 2, line 949, automated is doubled, typos on line 951, 952, 955

-All typos have been fixed.

- Double check paragraphs starting at line 978, 984.

- On lines 978 and 984, different categories of AAL architectures are presented in an itemized list.

- Line 1026 "CPS systems" (systems is redundant?)

Indeed, it was redundant and it was removed.

Round 2

Reviewer 1 Report

Dear Author,

I am pleased that you have taken my advice into considerations.

Thank you for the improvements you have introduced.

But, to be honest, I still haven't clearly understood the real usefulness of your approach.

In particular, I cannot see a situation in which the decision-supporting tools that you propose give a clear and significant contribution.

In the Introduction, you write: "Such integration is extremely beneficial in safety-critical situations, for instance, the case of a fall event occurring due to low pulse, which should trigger sending timely alerts to caregivers, for immediate intervention or else the life of the elderly can be endangered. This requires timely integration of health monitoring (in this case, pulse monitoring) and fall detection functionalities."

In my opinion, this example is not sufficient to explain the need for using formal techniques to analyze the integration of AAL Systems in order to guarantee the meeting of requirements.

I would have preferred a better example to help the reader understand where the problem is.

It may be that I am not sufficiently experienced in the specific sector of "formal verification".

However, I can guess the difficulty of modeling an AAL system in a formal way to verify its correct functioning. From this point of view, the work done is remarkable and well documented.

Author Response

Dear Reviewer,

Thanks again for your valuable comments. I have tried to explain the purpose of DSS more clearly now.

Review:

I am pleased that you have taken my advice into considerations.

Thank you for the improvements you have introduced.

Thanks a lot for the remark. Your advice was really helpful to improve our work.

But, to be honest, I still haven't clearly understood the real usefulness of your approach.In particular, I cannot see a situation in which the decision-supporting tools that you propose give a clear and significant contribution.In the Introduction, you write: "Such integration is extremely beneficial in safety-critical situations, for instance, the case of a fall event occurring due to low pulse, which should trigger sending timely alerts to caregivers, for immediate intervention or else the life of the elderly can be endangered. This requires timely integration of health monitoring (in this case, pulse monitoring) and fall detection functionalities."In my opinion, this example is not sufficient to explain the need for using formal techniques to analyze the integration of AAL Systems in order to guarantee the meeting of requirements.I would have preferred a better example to help the reader understand where the problem is.

Thanks for the valuable comment. To address the issue, we have now elaborated 3 cases in the Introduction where the integration of critical functionalities is crucial (Line 29-43). Also in Section 3.1 (line 352-418),  we explain the various use case scenarios and system requirements which again highlights the importance of integration.

 It may be that I am not sufficiently experienced in the specific sector of "formal verification".

However, I can guess the difficulty of modeling an AAL system in a formal way to verify its correct functioning. From this point of view, the work done is remarkable and well documented.

Thanks a lot for the appreciation.